# A multicentric consortium study demonstrates that dimethylarginine dimethylaminohydrolase 2 is not a dimethylarginine dimethylaminohydrolase

Vinitha N. Ragavan [1,2,22], Pramod C. Nair [2,3,4,5,22], Natalia Jarzebska[1], Ramcharan Singh Angom [6], Luana Ruta[7], Elisa Bianconi [7], Silvia Grottelli[8], Natalia D. Tararova[9], Daniel Ryazanskiy[9], Steven R. Lentz[10], Sara Tommasi[2], Jens Martens-Lobenhoffer [11], Toshiko Suzuki-Yamamoto[12], Masumi Kimoto[12], Elena Rubets[1], Sarah Chau[13], Yingjie Chen[14], Xinli Hu [15], Nadine Bernhardt [16], Peter M. Spieth[17], Norbert Weiss[1], Stefan R. Bornstein[1,18], Debabrata Mukhopadhyay [6], Stefanie M. Bode-Böger[11], Renke Maas [19,20], Ying Wang [13], Antonio Macchiarulo[7], Arduino A. Mangoni[2], Barbara Cellini[8] & Roman N. Rodionov [1,21] ✉

Dimethylarginine dimethylaminohydrolase 1 (DDAH1) protects against cardiovascular disease by metabolising the risk factor asymmetric dimethylarginine (ADMA). However, the question whether the second DDAH isoform, DDAH2, directly metabolises ADMA has remained unanswered. Consequently, it is still unclear if DDAH2 may be a potential target for ADMA-lowering therapies or if drug development efforts should focus on DDAH2's known physiological functions in mitochondrial fission, angiogenesis, vascular remodelling, insulin secretion, and immune responses. Here, an international consortium of research groups set out to address this question using in silico, in vitro, cell culture, and murine models. The findings uniformly demonstrate that DDAH2 is incapable of metabolising ADMA, thus resolving a 20-year controversy and providing a starting point for the investigation of alternative, ADMA-independent functions of DDAH2.

Nitric oxide (NO) is a key signalling molecule involved in regulation of vascular tone, inflammation, neuronal function, angiogenesis, and many other physiological and pathophysiological processes[1-4]. Asymmetric dimethylarginine (ADMA) is an endogenous homologue of L-arginine that inhibits NO production by all three known isoforms of NO synthase[5,6]. In addition to inhibition of NO production, ADMA also "uncouples" NO synthases, which results in production of superoxide radical instead of NO[7]. Multiple epidemiological studies have identified ADMA as an independent risk factor for cardiovascular and overall mortality in the general population and in patients with diseases of the

cardiovascular, pulmonary, renal, endocrine, and gastrointestinal systems[8-10]. Lowering ADMA in animal models resulted in protection against atherosclerosis, adverse myocardial and vascular remodelling, myocardial and renal ischaemia/reperfusion damage, and insulin resistance[11-15]. Elevation of ADMA, on the other hand, was shown to protect against tumour growth and septic shock[16,17].

Dimethylarginine dimethylaminohydrolase 1 (DDAH1) is the major enzyme responsible for metabolism of ADMA. It was first isolated from rat kidney and found to hydrolyse ADMA into citrulline and dimethylamine[18]. Overexpression of *Ddah1* depletes ADMA from

plasma and tissues, increases NO bioavailability, and protects from cardiovascular and metabolic injury in murine models[19–21]. In contrast, knockout of *Ddah1* in mice leads to elevated levels of ADMA in tissues and plasma and predisposes to cardiovascular injury[22]. In 1999, a second DDAH isoform, DDAH2, was identified[23]. DDAH1 and DDAH2 in humans are encoded by different genes localised on chromosomes 1p22 and 6p21.3, respectively[24]. The human DDAH1 and DDAH2 proteins have 50% identity in their amino acid sequences[25–27]. DDAH1 and DDAH2 are widely expressed in multiple tissues and cell types with partially overlapping and partially distinctive expression patterns; both genes are expressed at high levels in endothelial cells, kidney, and liver, with DDAH1 being more abundant in the brain and DDAH2 more abundant in the heart, lungs, and placenta[23,24,28].

There is no doubt that DDAH1 is an important and powerful regulator of ADMA homoeostasis[29,30]. The same cannot be said of DDAH2, however. The role of DDAH2 in ADMA metabolism has been a matter of debate for more than 20 years due to multiple contradictory reports regarding its enzymatic activity towards ADMA. On one hand, elevated levels of ADMA were detected in at least some tissues from homozygous *Ddah2* knockout mice[31,32], and lentivirus-mediated overexpression of *Ddah2* was reported to increase total DDAH activity in myocardium and reduce plasma ADMA levels in rats with streptozotocin-induced diabetes mellitus type 1[33]. On the other hand, there is considerable evidence questioning the role of DDAH2 in ADMA metabolism. For example, no DDAH activity was detected in porcine tissues that express DDAH2 but not DDAH1, such as thyroid gland[34]. Consistent with this observation, global *Ddah1* knockout mice had no detectable DDAH activity in all the examined tissues, including tissues with the highest levels of DDAH2 expression[35]. These findings suggested that either DDAH2 does not metabolise ADMA at all, or its role in ADMA metabolism is negligible. It was also reported that downregulation of DDAH2 (in contrast to downregulation of DDAH1) in rats by small interfering RNA did not result in an increase in serum ADMA levels[36]. Attempts to directly assess the enzymatic activity of DDAH2 towards ADMA have been limited by technical challenges in expressing and purifying DDAH2[34].

There is a growing body of evidence that DDAH2 plays a role in vascular and metabolic homoeostasis through mechanisms that are distinct from DDAH1 and ADMA. For example, DDAH2 induces expression of vascular endothelial growth factor (VEGF) via phosphorylation of transcription factor specificity protein 1 (Sp1) in a process that is independent of NO and NO synthase[37,38]. Similarly, overexpression of DDAH2 in mice led to Sp1-mediated transcriptional upregulation of secretagogin, an insulin vesicle docking protein[39]. DDAH2 has been also implicated in the regulation of gestational diabetes mellitus through Krupple-like factor 9[40]. This phenotype may be related to the observation that a polymorphism of *DDAH2* (at SNP rs2272592) is associated with the development of type 2 diabetes (T2D)[41]. Finally, recent evidence suggests a potential role for DDAH2 in regulating innate immune responses[42].

Since therapeutic approaches to lower ADMA in patients are predicted to result in major cardiovascular and metabolic benefits[22], it is important from a drug development perspective to define specific drug target(s) to achieve this goal. If both DDAH isoforms (DDAH1 and DDAH2) metabolise ADMA, then they should both be considered as targets for development of ADMA-lowering drugs. Alternatively, if DDAH2 does not metabolise ADMA, then development of ADMA-lowering therapies should be targeted towards DDAH1, while drug development targeting DDAH2 should be redirected towards its ADMA-independent biological effects in regulation of angiogenesis, diabetes, and immune responses.

The goal of the current study was to resolve the 20-year-old controversy regarding the involvement of DDAH2 in metabolism of ADMA. To address this question comprehensively, we assembled an international consortium of research groups working in the ADMA/ DDAH field and applied multiple complementary approaches. First, we used both static and dynamic modelling of the three-dimensional (3D) structure of DDAH2 to interrogate its ability to interact with ADMA in silico. Next, by using GST-tagged expression constructs, we succeeded in the notoriously challenging task of expressing and purifying soluble recombinant DDAH2 and investigated its ability to bind to ADMA and hydrolyse it to citrulline in vitro. Finally, we used multiple cell culture and in vivo models of deficiency of DDAH2 or DDAH1 to investigate the ability of DDAH2 to metabolise ADMA in human cells and mouse tissues.

All the complementary experiments performed by our multi-centre international consortium uniformly demonstrated that DDAH2, in contrast to DDAH1, is unable to hydrolyse ADMA to citrulline.

## Results

### Molecular docking/dynamics of ADMA in the 3D structure of DDAH2 demonstrates that ADMA does not fit into the predicted active site of DDAH2

We started our investigation of the role of DDAH2 in ADMA metabolism by examining the interactions between ADMA and the predicted active site of human DDAH2 in silico. Since DDAH 1 and 2 have high sequence similarity (~67%) and identity (~50%) (Supplementary Fig. 1–Needle Pairwise Sequence Alignment tool: EMBOSS Programs)[25–27], the DDAH1 X-ray crystal structure (PDB ID: 2JAI)[43] was used as a template to model the 3D structure of DDAH2 using SWISS-MODEL (Model A). We also used the artificial intelligence-derived AlphaFold DDAH2 structure (Model B) in our study. Both models were found to be nearly identical with a root mean square deviation (RMSD) of 0.78 Å, calculated by PyMol[44] (Fig. 1). Like the structure of DDAH1, the predicted structures of DDAH2 (Model A and B) comprise a barrel formed by five modules, each based on a similar ββαβ structural motif, with the predicted active site being in the middle of this barrel[43,45]. Structural overlay of DDAH1 and DDAH2 shows a highly similar overall fold (backbone RMSD, DDAH1 and Model A = 0.12 Å; DDAH1 and Model B = 0.76 Å (PyMol[44])) between the two isoforms (Fig. 1a). As Model A was built using the DDAH1 X-ray structure template, the positioning of flexible regions connecting regular secondary structures (e.g., loops) were nearly identical, unlike in Model B (Fig. 1a).

However, our analysis discovered a major difference between the active site of human DDAH1 and the predicted active site of human DDAH2. Specifically, the key catalytic residue Cys273 of DDAH1 is replaced by hydrophobic Leu 275 in DDAH2 Model A and by Ser274 in DDAH2 Model B (Fig. 1b–d). Residues Ser274 and Leu275 reside at a loop region, suggesting their adaptable positioning, which was subsequently confirmed in molecular dynamics simulations (see Discussion).

Molecular docking of ADMA in the DDAH1 binding site showed that the binding mode of ADMA is nearly identical to the orientation of the reaction product citrulline in the X-ray structure of the enzyme-product complex (PDB ID: 2JAI)[43]. By contrast, molecular docking of ADMA in the predicted (based on the DDAH1 template) substrate binding site of DDAH2 Model A showed that ADMA can only interact with residues outside of this putative substrate binding site, according to the top scored solution (Supplementary Fig. 2a, b). Furthermore, molecular docking of ADMA in the AlphaFold generated DDAH2 structure (Model B) did not result in any suitable ligand binding solution at the putative binding site. Blind docking was performed on this structure using a different programme (Flare, V6.1), which resulted in a binding pose similar to the one in Model A with N-dimethyl group (site of breakdown) facing the solvent exposed area, away from the putative binding site (Supplementary Fig. 2c). Next, we performed molecular dynamics simulation to gain further structural insights into differences in the binding of ADMA to human DDAH1 and DDAH2, taking into account protein flexibility. Molecular dynamics simulations showed that ADMA formed a stable interaction with the binding site of

DDAH1 with a low RMSD of 1.53 ± 0.24 Å (mean ± SD, Fig. 2 and Supplementary Table 1), showing some conformational flexibility at the dimethyl end of ADMA (Fig. 2a and Supplementary Movie 1). To investigate the possible binding mode of ADMA within the predicted DDAH2 binding site, ADMA was placed into the same position in the predicted DDAH2 structure (Model A) as citrulline in the DDAH1 X-ray structure. However, in contrast to DDAH1, molecular dynamics simulation showed that ADMA was almost immediately (<5 ns) displaced from its originally set position, thus suggesting that binding is not favoured (Fig. 2b–d and Supplementary Movie 2). Multiple subsequent molecular dynamics simulations confirmed the highly unstable binding mode of ADMA in the predicted substrate binding site of DDAH2 (Model A) (Fig. 2f and Supplementary Table 1−RMSD = 5.23 ± 1.99 (mean ± SD), $p = 7.7 \times 10^{-7}$ vs. RMSD of ADMA binding to DDAH1). Molecular dynamics simulations of ADMA bound to DDAH2 Model B showed a relatively longer residence time of ADMA near the predicted binding site, when compared with Model A, however, this interaction nevertheless also led to unstable binding (Fig. 2g and Supplementary Movie 3 and Supplementary Table 1−RMSD = 3.81 ± 0.88 (mean ± SD), $p = 2.2 \times 10^{-3}$ vs. RMSD of ADMA binding to DDAH1).

Overall, these in silico data confirm that ADMA is a favoured ligand of DDAH1; in contrast, the formation of a stable ADMA-DDAH2 complex productive for the hydrolytic reaction is highly unfavourable.

**Thermophoresis studies demonstrate stable binding of ADMA to recombinant human DDAH1 but unstable binding to DDAH2**

To directly investigate the DDAH1-ADMA and DDAH2-ADMA interaction, we performed studies with purified recombinant human DDAH1 and DDAH2 as N-terminal fusion proteins with glutathione S-transferase (GST) (DDAH1-GST and DDAH2-GST). Purified GST was used as a negative control to exclude nonspecific effects of the tag. We first performed MicroScale Thermophoresis (MST)[46] to investigate possible direct interactions between ADMA and DDAH1-GST and DDAH2-GST proteins. Thermophoresis is the movement of a biomolecular complex in a temperature gradient which depends on different parameters, such as size, charge, and hydration shell, that vary upon a ligand binding event to the target protein[47]. Briefly, the MST experiment is based on the use of 16 capillary tubes filled with a fluorescent dye-labelled protein and titrated with decreasing concentrations of unlabelled ligand. These tubes are then illuminated with an infrared laser that generates a temperature gradient. The protein/ligand complex migrates along this gradient, causing changes in the observed fluorescence that are used to generate a binding curve as a function of ligand concentration and calculate the dissociation constant ($K_d$).

We compared the ability of ADMA to bind to fluorescently labelled DDAH1-GST and DDAH2-GST. Both DDAH1-GST and DDAH2-GST were soluble. There was no indication of recombinant DDAH2 being improperly folded from its intrinsic fluorescence spectrum (Supplementary Fig. 3). A binding experiment between ADMA and fluorescently labelled recombinant purified GST was also carried out as negative control to rule out any interfering interaction of the substrate with the GST domain of the constructs. Figure 3 shows the titration curves of ADMA binding to DDAH1-GST, DDAH2-GST, and GST. A clear binding event was observed between ADMA and DDAH1-GST, yielding a $K_d$ of 4.27 ± 0.46 μM (Fig. 3 and Table 1) with both signal amplitude and signal-to-noise ratio being above the accepted thresholds

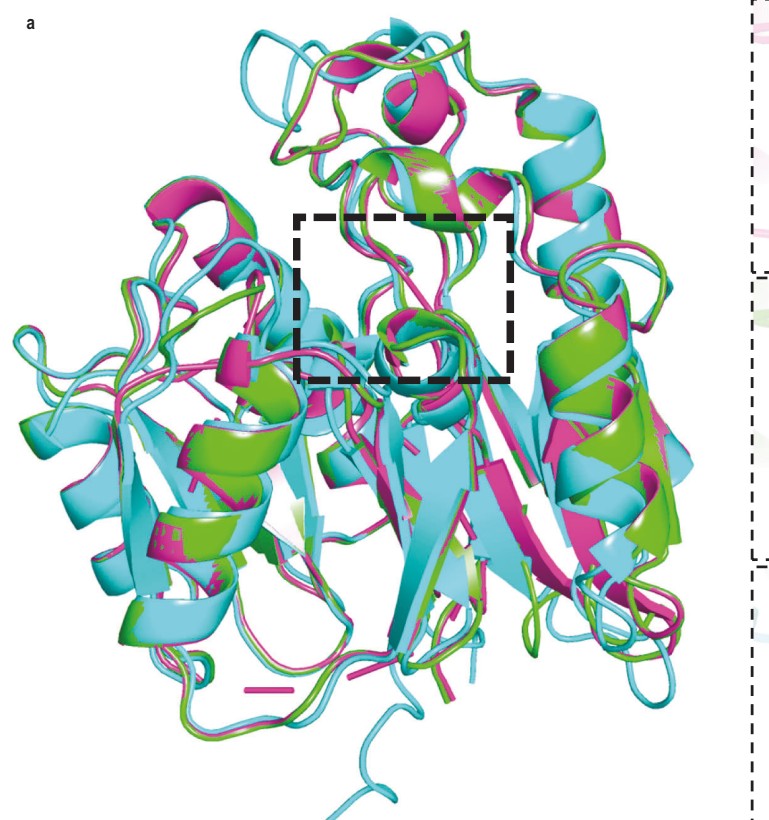
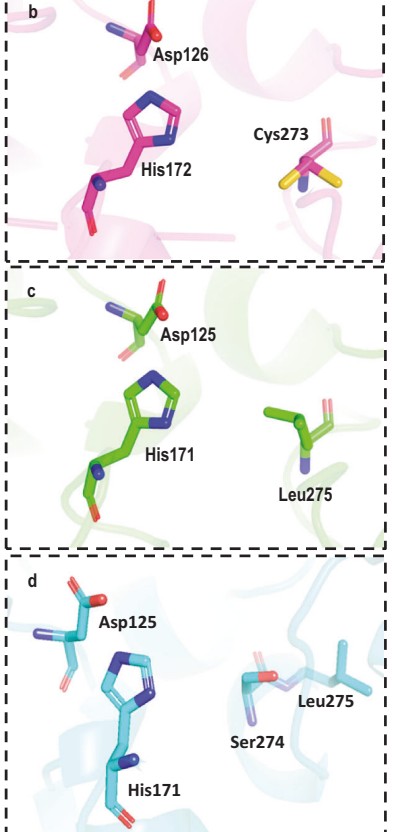

**Fig. 1 | Comparison of the 3D structures of DDAH1 and DDAH2. a** Structural overlay of DDAH1 X-ray structure 2JAI (magenta) and DDAH2 homology models (green, cyan) (cartoon). DDAH1 superimposed to DDAH2 Model A and B have a root mean square deviation (RMSD) of 0.12 Å and 0.76 Å, respectively, while the superimposed DDAH2 Model A to Model B has a RMSD of 0.78 Å (PyMol[44]).

**b** Catalytic triad of DDAH1 binding site (residue numbering from Leiper et al.[43]). **c** Residues at the equivalent position in DDAH2 site SWISS-MODEL (Model A, green). **d** Residues at the equivalent position in DDAH2 site AlphaFold (Model B, cyan).

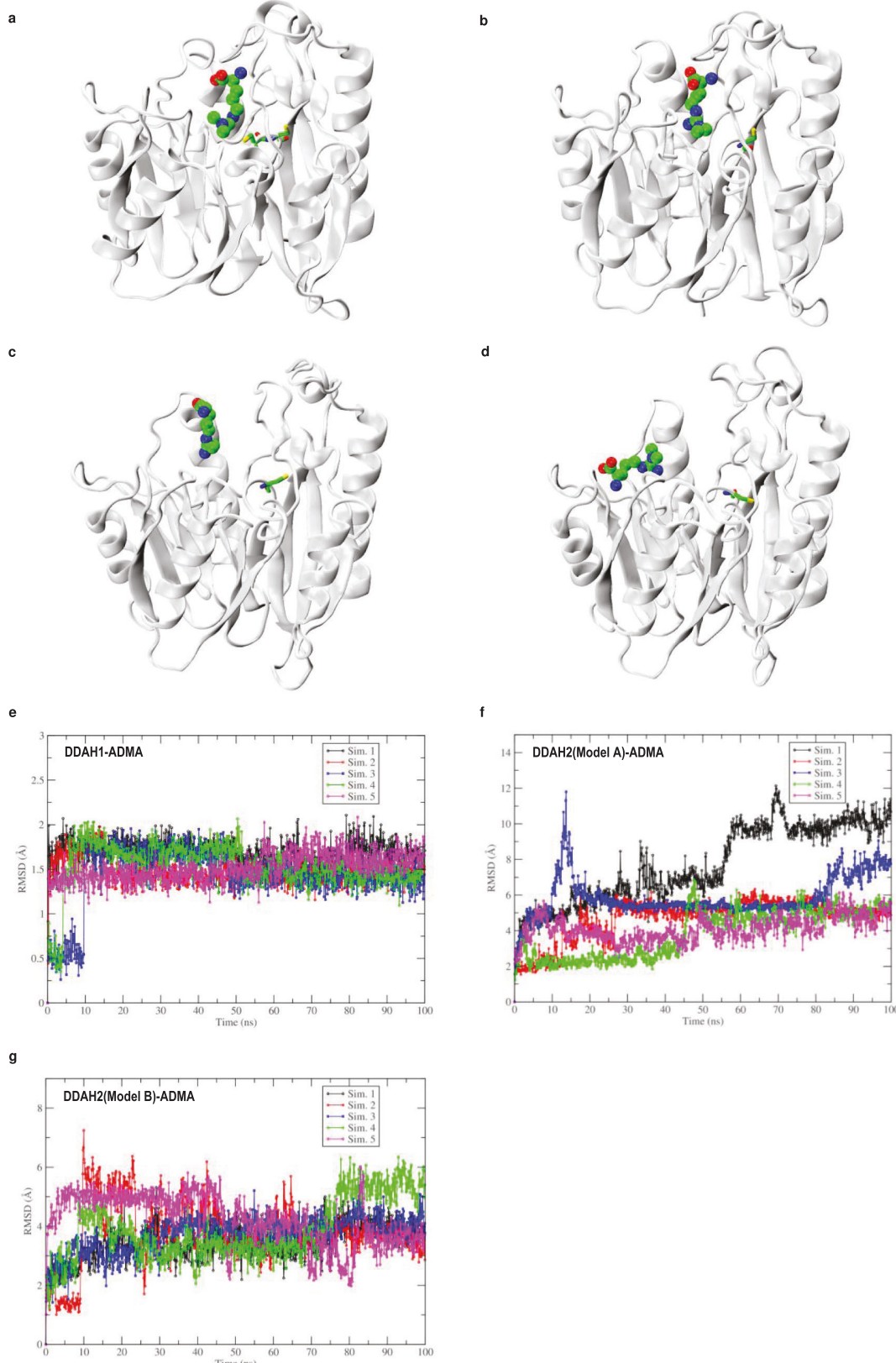

**Fig. 2 | Molecular dynamics study of DDAH proteins. a** Binding mode of ADMA in DDAH1-active site (average conformation) (cartoon), Cys273 and Cy274 are shown in sticks. Binding mode of ADMA in DDAH2 (Model A, cartoon) binding site at (**b**) 0 ns (start of simulations), (**c**) 40 ns, and (**d**) 100 ns. In DDAH2 structure, Cys276 (equivalent of Cy274 in DDAH1) is shown in sticks. Root mean square deviation (RMSD) of ADMA bound to DDAH proteins from five independent molecular dynamics simulations of individual DDAH-ADMA complexes (represented as Sim. 1–5), ADMA bound to (**e**) DDAH1, (**f**) DDAH2 (Model A), and (**g**) DDAH2 (Model B). Sim. simulation. Source data are provided as a Source Data file.

(Amplitude (Ampl) = 11.7; Threshold >8.4 and signal-to-noise ratio (S/N) = 30.9, threshold >5). Binding of ADMA to DDAH2-GST was also observed with a $K_d$ of 10.75 ± 3.28 μM (Fig. 3 and Table 1), however, variations in the amplitude (Ampl = 5.2) were below the accepted threshold value of >10.5, indicating an unstable and/or nonspecific interaction. This finding is consistent with the in silico molecular dynamics simulations. Finally, no binding was observed between ADMA and the GST protein domain, confirming the specificity of the observed binding events to the DDAH1 and DDAH2 domain of the constructs.

## Recombinant DDAH2 does not metabolise ADMA

To determine whether the unstable binding of ADMA to DDAH2 observed in the thermophoresis experiments can support catalytic hydrolysis of ADMA to citrulline, we investigated the ability of recombinant human DDAH2-GST to enzymatically convert ADMA to L-citrulline in vitro with DDAH1-GST serving as the positive control. As shown in Fig. 4, we were able to detect the ADMA-hydrolysing activity of recombinant DDAH1-GST, thus further confirming that the presence of the tag does not interfere with protein structural and functional features. On the other hand, no activity was detected for recombinant DDAH2-GST, even at a protein concentration 10-fold higher than that used in the DDAH1-GST sample. GST protein, which was used as an internal negative control, did not show any activity towards ADMA as well.

## DDAH2 overexpression has no measurable effect on ADMA metabolism

We proceeded with our investigation into the role of DDAH2 in ADMA metabolism by examining the effect of DDAH2 overexpression in cultured cells on ADMA conversion to citrulline. Lysates from HEK293T cells stably transfected with either *DDAH1* or *DDAH2* were incubated with a range of ADMA concentrations and the corresponding citrulline product was measured by ultra-performance liquid chromatography (UPLC) coupled to mass spectrometry. *DDAH1* overexpression in HEK293T cells resulted in increased conversion of ADMA to citrulline, while *DDAH2* overexpression did not have any effect on citrulline production from ADMA (Fig. 5).

## DDAH2 gene deletion does not affect ADMA metabolism in MDA-MB-231 cells

We next investigated the effect of *DDAH2* gene deletion on ADMA metabolism. MDA-MB-231 *DDAH* knockout cell lines were generated using the CRISPR/Cas9 system by inducing a deletion in either exon 3 of the coding sequence of *DDAH1* or in exon 1 of the coding sequence of *DDAH2*. Double knockout clones *(DDAH1 + DDAH2)* were generated by performing the *DDAH1* deletion in the DDAH2 knockout cell line. We confirmed the deletion of the target region for each of the clones by PCR genotyping using *DDAH1* and *DDAH2* specific primer pairs (Supplementary Fig. 4). We also demonstrated that the knockout clones did not have any detectable mRNA or protein expression for the targeted genes by quantitative PCR (qPCR) and Western blotting, respectively (Fig. 6a–c). When *DDAH2* knockout cells were incubated with D7-labelled ADMA, no detectable decrease in D7-labelled citrulline production was observed compared with wild-type cells, whereas deletion of *DDAH1* in either *DDAH1* knockout or double *(DDAH1 + DDAH2)* knockout clones resulted in a complete loss of the ability of cells to metabolise D7-labelled ADMA to D7-labelled citrulline (Fig. 6d).

## Downregulation of DDAH2 does not affect ADMA metabolism in cultured primary endothelial cells

We further evaluated the role of DDAHs in ADMA metabolism in human primary endothelial cell, HUVECs. *DDAH* knockdown HUVECs were generated by infection of lentivirus vectors which express control shRNA, *DDAH1* shRNA, or *DDAH2* shRNA. We confirmed the specific

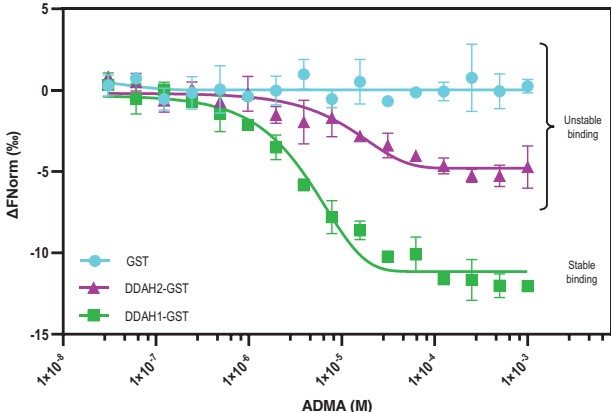

**Fig. 3 | Titration curves of ADMA to DDAH1-GST, DDAH2-GST, and GST as resulting from the MST binding experiments.** A stable binding event occurs only in the case of ADMA to DDAH1-GST (green). Conversely, variations in the amplitude and in the signal-to-noise ratio compared to background noise are below the accepted threshold values for the binding event of ADMA to DDAH2-GST (magenta). No binding event is obtained in assessing the interaction of ADMA to GST (cyan) protein domain. Data (*n* = 3 technical replicates) are presented as mean ± SD. Source data are provided as a Source Data file.

knockdown of *DDAH1* and *DDAH2* using both qPCR and Western blotting (Fig. 7a–c). As seen in Fig. 7d, DDAH activity remained unchanged in HUVECs treated with *DDAH2* shRNA.

## Homozygous Ddah1 knockout leads to complete loss in the total DDAH activity in mouse tissues

Next, we examined the role of DDAH2 in ADMA metabolism in vivo by investigating the residual tissue DDAH activity in mice with homozygous deletion of *Ddah1 (Ddah1$^{-/-}$)*. As expected, *Ddah1$^{-/-}$* mice did not have any detectable DDAH1 expression in all examined tissues (heart, lungs, liver, kidney, brain), while the levels of DDAH2 expression remained unaffected (Fig. 8a–c). Interestingly, homozygous deletion of *Ddah1* led to complete loss of total detectable tissue DDAH activity, as measured by conversion of D7-labelled ADMA to D7-labelled citrulline, despite preserved DDAH2 expression (Fig. 8d).

## Homozygous Ddah2 knockout does not affect total DDAH activity in mouse tissues

Next, we examined the effect of global *Ddah2* deletion on tissue DDAH activity. The *Ddah2$^{-/-}$* mice had no detectable DDAH2 mRNA or protein expression (Fig. 9a–c). We did not observe any changes in tissue DDAH activity in *Ddah2$^{-/-}$* mice compared to the wild-type littermates (Fig. 9d).

## Discussion

The main findings of our study are that (1) according to molecular docking and molecular dynamics studies, ADMA is unable to bind efficiently to the predicted substrate binding site of DDAH2; (2) binding of ADMA to recombinant human DDAH2 detected in thermophoresis analysis is suggestive of ADMA binding modes that are catalytically unproductive, transient, and unstable as noted in the simulations; (3) recombinant human DDAH2, in contrast to recombinant human DDAH1, is unable to metabolise ADMA to citrulline; (4) overexpression of human *DDAH2* in cultured human cell line HEK293T, in contrast to overexpression of human *DDAH1*, does not affect total DDAH activity (conversion of ADMA to citrulline) in cell lysates; (5) knockout of DDAH2 in cultured human cell lines does not affect total DDAH activity, whereas knockout of DDAH1 in the same cell lines results in complete loss of total DDAH activity; (6) downregulation of DDAH2 in human primary endothelial cells does not affect total DDAH

**Table 1 | Dissociation constants ($K_d$) and confidence values ($K_d$ ± confidence values) of ADMA binding to DDAH1-GST, DDAH2-GST, and GST with statistical parameters of result's significance (amplitude, signal to noise ratio (S/N))**

| Binding experiment | Amplitude (threshold value) | S/N (threshold value) | Binding status $K_d$ (µM) ± confidence values |
|---|---|---|---|
| ADMA/DDAH1-GST | 11.7 (8.4) | 30.9 (5) | Stable binding 4.27 ± 0.46 |
| ADMA/DDAH2-GST | 5.2 (10.5) | 10.6 (5) | n.s Amplitude < threshold |
| ADMA/GST | 0.2 (2.7) | 0.5 (5) | n.s Amplitude < threshold |

*n.s.* non-significant.

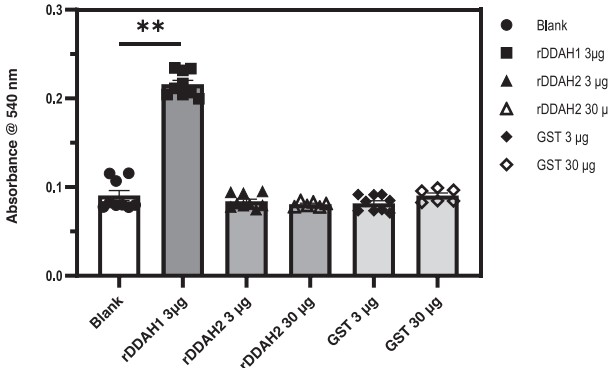

**Fig. 4 | L-citrulline formation by recombinant DDAH.** Recombinant DDAH and GST proteins were incubated with 10 mM ADMA, and the enzymatic activity was measured by detection of L-citrulline production. The DDAH specific enzymatic activity of rDDAH1 was 17.10 ± 0.4 (pmoles/min/µg ± SD). The experiments with recombinant DDAH2 30 µg and GST 30 µg were performed twice while the remaining experiments with the other conditions were performed three times. Kruskal–Wallis test with Dunn's multiple comparison test to the blank was performed. $n = 9$ for Blank, 3 µg GST protein, 3 µg rDDAH1, 3 µg rDDAH2; $n = 6$ for 30 µg GST protein, 30 µg rDDAH2 (technical replicates). Data are presented as mean ± SD. **$p < 0.01$ ($p = 0.0035$). Source data are provided as a Source Data file.

activity; (7) global homozygous deletion of *Ddah1* in mice results in the complete loss of total DDAH activity in tissue lysates; and (8) homozygous loss of *Ddah2* expression in mice does not affect total DDAH activity in tissue lysates.

The identification of a putative second DDAH isoform by Leiper et al.[23] heightened interest in ADMA metabolism and the biological role of DDAH2. Based on the similarity in the amino acid sequences of DDAH1 and DDAH2 (Supplementary Fig. 1), it was proposed that DDAH2 might have similar enzymatic activity towards ADMA as DDAH1[23]. However, subsequent studies yielded contradictory results regarding the enzymatic activity of DDAH2 towards ADMA and led to a major controversy in this research field[34,35,48,49], which prompted our interest towards resolving this question in a definitive multicentre study.

We started our study by first examining the 3D structure of human DDAH2 protein. Since the crystal structure of DDAH2 has not been resolved, we used the known X-ray crystal structure of human DDAH1 to generate a homology model of human DDAH2 (Model A) and also examined the AlphaFold derived structure (Model B). The X-ray structure of human DDAH1 shows two cysteines, Cys273 and Cys274, within the active site, with the first one involved in the catalytic cycle[50–52]. This structure is consistent with the function of DDAH1 as an aminohydrolase that catalyses the metabolism of its substrates (e.g., ADMA) via a catalytic triad of Cys273, His172, and Asp126 (amino acid numbering based on Leiper et al.[43], Fig. 1b), with a mechanism involving nucleophilic attack on the substrate by the Cys273 residue. In contrast to DDAH1, DDAH2 has only one cysteine, Cys276 (equivalent of Cys274 in DDAH1) in the predicted substrate binding site. Notably, the 3D structural alignment of DDAH1 and DDAH2 shows that Cys273 in the catalytic triad of DDAH1 is replaced by either Leu275 (Model A)

or Ser274 (Model B) in predicted DDAH2 structures (Fig. 1c, d). As noted earlier, the residues Ser274 and Leu275 reside at the loop region and molecular dynamics simulations show flexible positioning of these residues (Supplementary Movie 4), suggesting that multiple conformational states of this loop region may be available during the ligand binding process. However, in either case, the hydrophobic Leu275 is unable to perform nucleophilic attack on the substrate ADMA. Further, it was demonstrated in *Pseudomonas aeruginosa* DDAH (*Pa* DDAH) that mutation of catalytic Cys273 to Ser results in a failure to catabolise ADMA[45]. Thus, neither Ser274, nor Leu275 is capable of supporting the catalytic breakdown of ADMA. Finally, as demonstrated by the molecular docking analysis, ADMA is unable to bind stably to the predicted active site of DDAH2 in the same orientation as it binds to the active site of DDAH1.

Molecular dynamics simulations of ADMA in the human DDAH1 3D structure showed a very stable binding mode with some flexibility near the methyl groups of ADMA, which is consistent with the X-ray structure of *Pa* DDAH in the presence of ADMA, where the high heat capacity of two methyl groups in comparison to whole ADMA molecule were detected[45]. Importantly, the binding mode of ADMA within the DDAH1 binding site was nearly identical to the binding mode of citrulline in the previously published structure of human DDAH1 (PDB ID: 2JAI)[43]. Thus, our molecular dynamics simulations were able to correctly reproduce the ADMA conformation to the experimentally known conformation of its metabolite citrulline as noted in the human DDAH1 structure (Fig. 1a and Supplementary Movie 1), thereby confirming that ADMA binds favourably within the DDAH1 binding site. In contrast, ADMA did not form any stable interactions with DDAH2 in either model (Fig. 2b–d and Supplementary Movie 2 and 3). This was confirmed by multiple simulations performed in a variety of alternative binding modes (Supplementary Fig. 5). Irrespective of its initial placement, ADMA was ejected from the putative binding pocket of DDAH2 Model A within 5 ns of simulation. In Model B, ADMA showed high flexibility (positioning in different binding modes) in the binding site before the unbinding event. For example, in Sim. 2 and Sim. 4 of Model B (Fig. 2g), the unbinding is observed beyond 10 ns after the initial phase of the unstable binding orientation of ADMA. The simulations of ADMA in Model B show interaction with Ser274 with an extended residence time compared to Model A. This is consistent with the previously reported data on the *Pa* DDAH structure, whereby ADMA formed an equivalent interaction within the Cys273Ser DDAH protein despite its inability to catabolise the substrate[45]. We also performed simulations to monitor the positioning of ADMA with respect to Cys276 to determine if this residue of DDAH2 could support nucleophilic attack and catalysis and found that ADMA was not at a catalytically favourable distance from Cys276 (Fig. 2b–d). Furthermore, a significantly higher RMSD of ADMA is observed in both DDAH2 models (Model A and B) when compared to the DDAH1 structure (Fig. 2e–g). Overall, the data from the molecular docking calculations and molecular dynamics simulations support each other and indicate that binding of ADMA to DDAH2 is unstable, and that there is no required local environment within this region to catalyse ADMA hydrolysis to citrulline.

A strength of our study is that we used purified human recombinant DDAH proteins to directly investigate their ability to bind ADMA

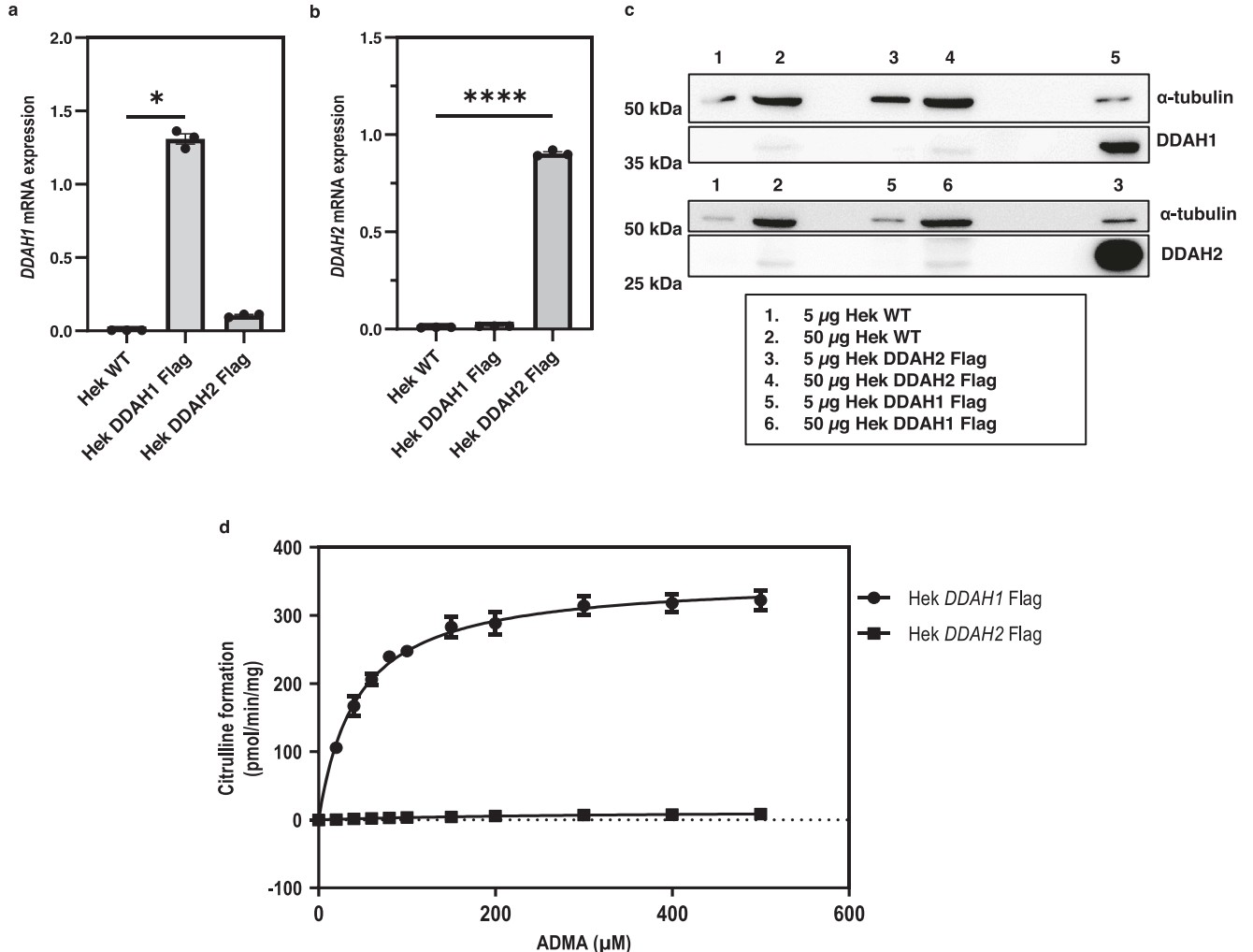

**Fig. 5 | ADMA metabolism by HEK293T cells overexpressing *DDAH1* and *DDAH2*.**
**a**–**c** mRNA and protein expression of HEK293T cells stably transfected with *DDAH1* or *DDAH2* in comparison to wild-type cells. **d** HEK293T cells with *DDAH1* or *DDAH2* overexpression were used to measure citrulline formation, a product of ADMA metabolism, at increasing substrate concentrations. Kruskal–Wallis test with Dunn's multiple comparison test to the wild-type cell line was performed for *DDAH1* mRNA expression analysis while one-way ANOVA with multiple comparisons to the wild-type cell line was performed for *DDAH2* mRNA expression analysis ($n = 3$ biological replicates). DDAH activity assay using ADMA performed with $n = 3$ and $n = 2$ (biological replicates) respectively for *DDAH1* and *DDAH2* HEK293T cell models. Data are presented as mean ± SE. **a** *$p < 0.001$ ($p = 0.0146$); **b** ****$p < 0.0001$ ($p < 0.000001$). Source data are provided as a Source Data file.

and hydrolyse it to citrulline. Consistent with our in silico analysis, we were able to detect only unstable binding of ADMA to recombinant DDAH2 in thermophoresis studies. Also, consistent with our molecular docking calculations, only recombinant human DDAH1, but not DDAH2, was able to catabolise ADMA hydrolysis to citrulline (Fig. 4). In a prior study, recombinant human DDAH2 protein expressed by *Escherichia coli* cells was found to metabolise both ADMA and L-NMMA[23]. However, unlike our current study, the recombinant human DDAH2 protein used in that study was not purified, which raises the possibility that the detected activity in bacterial lysates might have come not from the recombinantly expressed human DDAH2, but from the endogenous bacterial DDAH enzyme.

Our study showed that overexpression of human DDAH2 in cultured human cells, in contrast to human DDAH1, did not affect total DDAH activity in the cell lysates (Fig. 5). In a complementary experiment, DDAH2 knockout in a cultured human cell line did not affect total DDAH activity, while DDAH1 knockout resulted in complete loss of DDAH activity in the cell lysates (Fig. 6). We confirmed these effects in primary human endothelial cells (Fig. 7). Our data are consistent with previous observations in bovine aortic endothelial cells, in which knockdown of DDAH2, in contrast to knockdown of

DDAH1, did not affect total DDAH activity[53]. As acknowledged in the introduction section of this manuscript, the results of previous cell culture studies investigating the effect of DDAH2 expression modulation on ADMA metabolism are contradictory; however, an advantage of our studies and the experiments in bovine aortic endothelial cells performed by Pope and colleagues is that we measured DDAH activity in cell lysates using a highly sensitive and reliable stable isotope-based method, whereas most of the prior cell culture studies simply measured ADMA concentrations in culture medium or cell lysates[31,54,55]. Our results, however, contradict the previous observation that DDAH2 overexpression increased total DDAH activity in microvascular endothelial cells, where DDAH activity was also measured by an isotope-based method[56]. There is a similar discrepancy regarding the possible effect of DDAH2 over-expression on total tissue DDAH activity in vivo, with some of the studies reporting increased total DDAH activity in the tissues with DDAH2 overexpression[33,48,56]. Taking into account that our analysis of 3D structure of DDAH2 showed that ADMA cannot serve as a substrate for this enzyme, a possible explanation of why overexpression of DDAH2 in certain studies affects total DDAH activity, might be putative stimulating effects of DDAH2 upregulation on the enzymatic

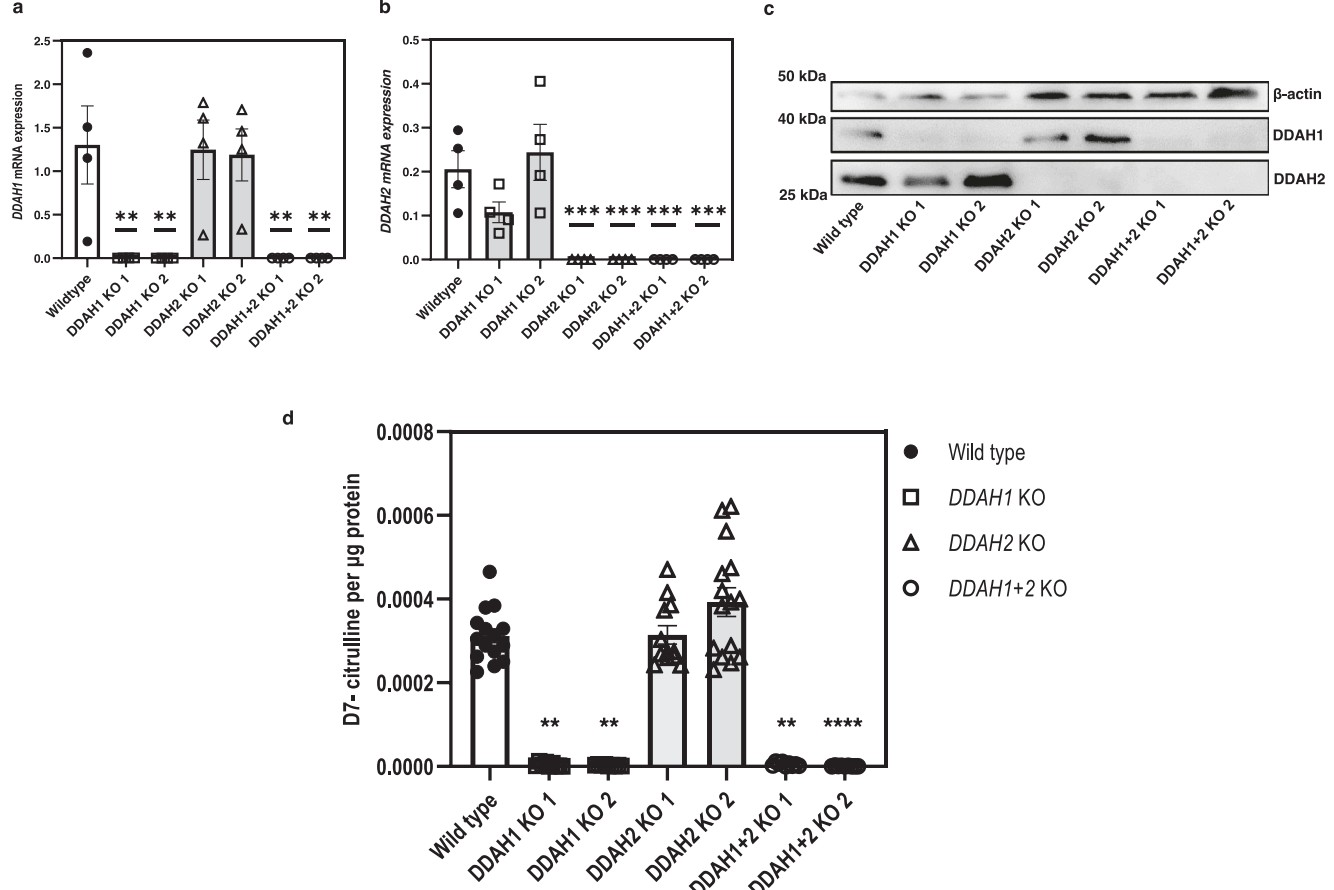

**Fig. 6 | DDAH activity assay of lysates from wild type and *DDAH* knockout MDA-MB-231 cells using D7-labelled ADMA as substrate. a**, **b** mRNA expression of both *DDAH1* and *DDAH2*, respectively, in the cell lines. **c** Representative Western blot image of MDA-MB-231 wild-type cells and the respective *DDAH* knockout clones. **d** Levels of D7-labelled citrulline produced by metabolism of D7-labelled ADMA. Data from the *DDAH1* and *DDAH2* mRNA expression analyses were collected from four experiments ($n = 4$ biological replicates) and one-way ANOVA statistical analyses with comparison to the wild-type group was performed for the normally distributed data. Data from the DDAH activity assay was collected from at least three experiments, performed in triplicate (MDA-MB-231 wild-type, DDAH2 KO 1, and DDAH2 KO 2 $n = 15$; DDAH1 KO 1, DDAH1 KO 2, DDAH1 + 2 KO 1, and DDAH1 + 2 KO 2 $n = 9$, biological replicates). Statistical analysis was performed by Kruskal–Wallis test for non-normally distributed data, with Dunn's multiple comparison to the wild-type cell line. Outliers were removed based on Grubbs (DDAH2 KO 1 $n = 3$). Data are presented as mean ± SE. **a** $**p < 0.01$ ($p = 0.006$); **b** $***p < 0.001$ ($p = 0.00057$); **d** $**p < 0.01$ (DDAH1 KO 1 $p = 0.0013$, DDAH1 KO 2 $p = 0.0014$, DDAH1 + 2 KO 1 $p = 0.0078$), $****<0.0001$ ($p = 0.000007$). Source data are provided as a Source Data file.

activity of DDAH1, e.g. by scavenging some endogenous negative regulator of DDAH1.

A major strength of our study is that we investigated the effect of DDAH1 and DDAH2 deficiency on tissue DDAH activity in mice. In agreement with our cell culture data, no residual DDAH activity could be detected in the tissues from homozygous *Ddah1* knockout mice and, using rigorous assay conditions, we could not detect any decrease in the total DDAH activity in the tissues of mice with homozygous *Ddah2* knockout. Our results are consistent with previous observations of complete loss of the total tissue DDAH activity in mice with homozygous *Ddah1* deficiency[35], as well as with the observation that heterozygous *Ddah1* deficiency results in about 50% decrease in the total tissue DDAH activity[43]. However, our finding that total DDAH activity in mouse tissues was unchanged in *Ddah2* homozygous knockout mice contrasts with a previous report of elevated ADMA concentrations in certain tissues from mice with homozygous *Ddah2* deficiency[31], yet, there are multiple alternative mechanisms for elevation of tissue ADMA concentration in addition to changes in total DDAH activity, such as changes in ADMA production or transport, or in case of the kidneys and liver also changes in ADMA metabolism by alanine:glyoxylate aminotransferase 2 (AGXT2)[57–60]. Our findings, however, are consistent with the previous observations that no DDAH activity could be detected in those porcine tissues, which expressed only DDAH2, but no DDAH1[34].

In summary, our consortium study provides the multiple complementary levels of evidence that, in contrast to DDAH1, DDAH2 is unable to use ADMA as a substrate. While our study resolves the longstanding contradiction regarding the ability of DDAH2 to metabolise ADMA, we do not challenge the other reported physiological and pathophysiological effects of DDAH2. On the contrary, we believe that DDAH2 is a highly important regulator of angiogenesis, vascular permeability, immune response and cellular growth, proliferation, senescence, and death[31,38,61–63]. These effects are likely mediated by ADMA-independent effects of DDAH2 on NO metabolism[36], protein-protein interactions[38], effects on mitochondrial fission[42,64], and/or possible enzymatic effects on other substrates rather than ADMA, which have not yet been identified. Our study therefore opens a new research field, which will allow investigation and possible therapeutic targeting of ADMA-independent DDAH2-mediated biological effects.

## Methods
### Protein modelling and molecular docking calculations
The 285 amino acid sequence of DDAH2 was obtained from the Uni-Prot protein database (https://www.uniprot.org/) (UniProt entry

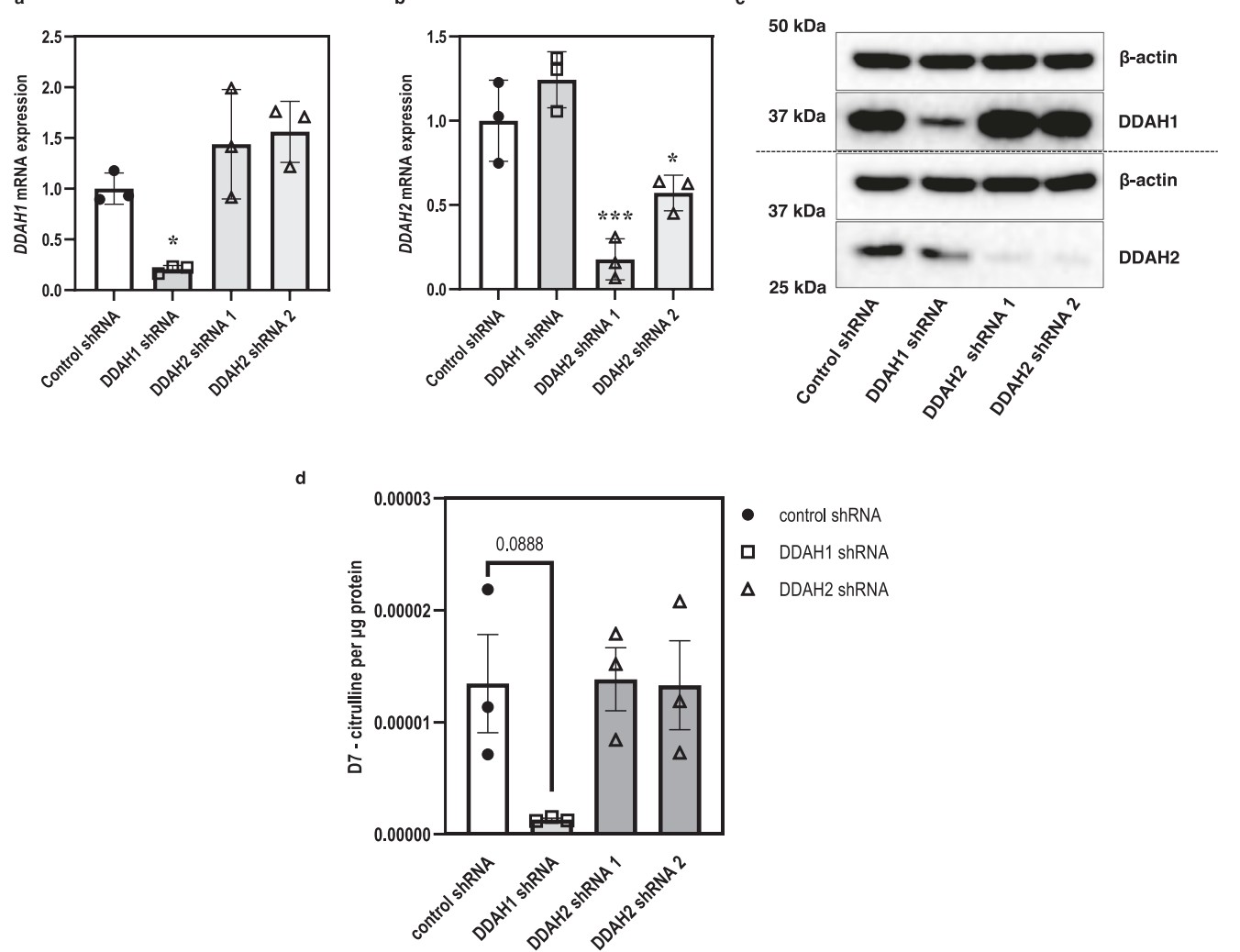

**Fig. 7 | DDAH activity assay of lysates from control and shRNA *DDAH* knock-down HUVECs using D7-labelled ADMA as substrate. a, b** mRNA expression of both DDAH1 and DDAH2, respectively, in the cell lines. **c** Representative Western blot image of control and DDAH shRNA knockdown clones. **d** Levels of D7-labelled citrulline produced by metabolism of D7-labelled ADMA. Statistical analysis was performed using one-way ANOVA with comparisons to the control shRNA group ($n = 3$ biological replicates). Data are presented as mean ± SE. **a** $*p < 0.05$ ($p = 0.0464$); **b** $*p < 0.05$ ($p = 0.0272$), $***p < 0.001$ ($p = 0.0009$). Source data are provided as a Source Data file.

O95865). The X-ray crystal structure of DDAH2 has yet to be generated. Therefore, the X-ray crystal structure of DDAH1 (PDB ID: 2JAI) chain A[43], was used to generate a homology model of the human DDAH2 structure using SWISS-MODEL[65]. The Alpha Fold DDAH2 structure (Model B) was obtained from https://alphafold.ebi.ac.uk/[66,67].

Molecular docking calculations were used to evaluate the binding mode and interactions of ADMA with DDAH1 or DDAH2 proteins. The crystal structure of DDAH1, 2JAI[43], was used as a template in the molecular docking studies. Protein structures were prepared by including hydrogen atoms (H-atoms) Kollman all-atoms charges using the BioPolymer module of SYBYL (version X-2.1, Certara, Princeton, NJ, USA). 3D coordinates of ADMA in structure data file (sdf) format were obtained from Pubchem (https://pubchem.ncbi.nlm.nih.gov/)[68], and molecular modelling was achieved using SYBYL, installed on a Red Hat Linux 6.9 OS workstation. After the assignment of Gasteiger–Huckel partial atomic charges[69], energy minimisation was performed using Powell's conjugate gradient method in conjunction with a Tripos 5.2 force field[70,71]. A minimum energy difference of 0.001 kcal/mol was set as the convergence criterion. Molecular docking experiments were conducted using the Surflex-Dock docking suite[72] as previously

reported[73] with the resulting binding poses ranked according to the total score (SYBYL Surflex-Dock). PyMol[44] was used to compare deviations between structure. Additional molecular docking experiments were performed using Flare (V6.1, Cresset®, Litlington, Cambridgeshire, UK; http://www.cresset-group.com/flare/).

**Molecular dynamics simulations**
Molecular dynamic simulations were performed using our previous approaches[74]. Specifically, the DDAH1/DDAH2 proteins bound to ADMA were simulated using GROMACS 2020 in conjunction with the AMBER14SB force field[75,76]. The transferable intermolecular potential with 3 points (TIP3P) water model was used to describe the solvent water. Simulations were performed under periodic boundary conditions in a rectangular box. The Lennard–Jones interactions were calculated with a 1.2 nm cut-off, whereas the electrostatic interactions were calculated using particle mesh Ewald summation. ADMA is predicted to have a charge state of +1 at pH 7.4, using the Calculator Plugins implemented in ChemAxon (Marvin 16.6.20). Topology parameters of ADMA were obtained according to GAFF (Generalised Amber Force Field) using the ACPYPE server[77]. ADMA was docked within the DDAH1 binding site using the reference position of the citrulline in the

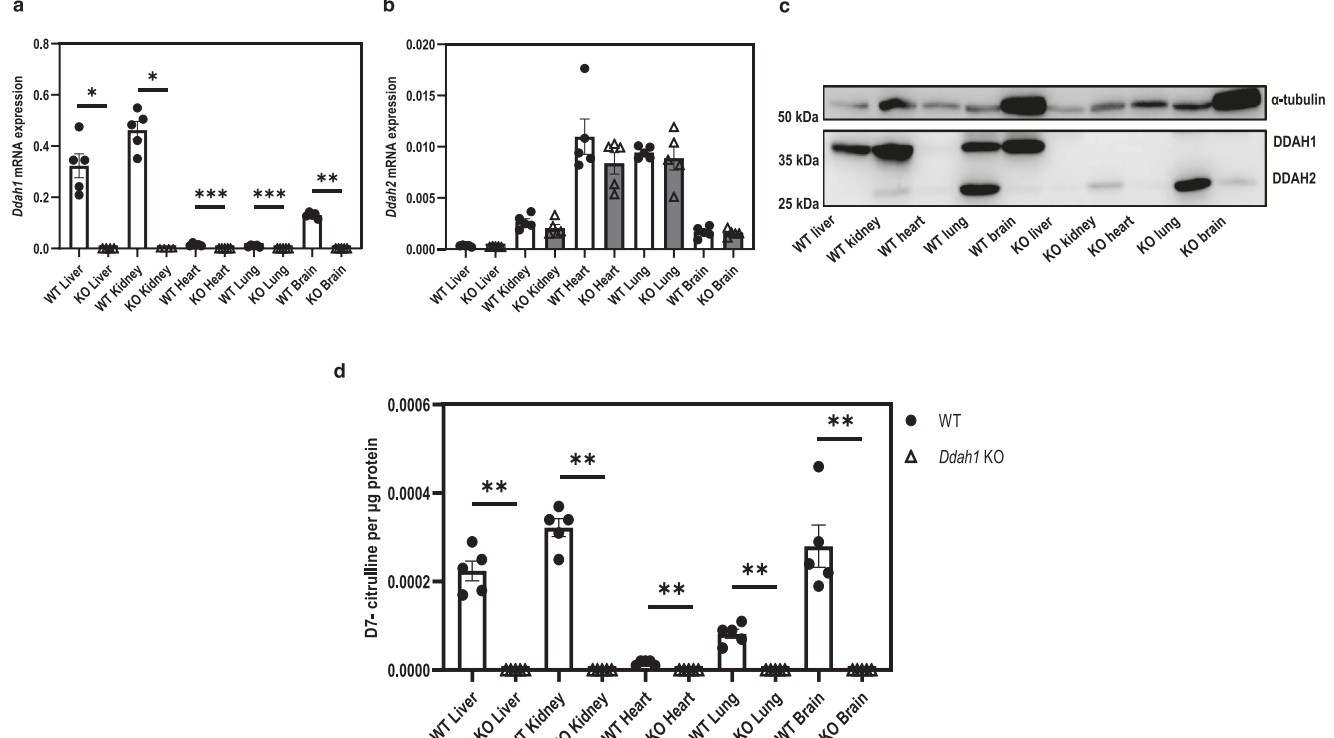

**Fig. 8 | DDAH activity assay of tissues from wild type and *Ddah1* knockout mice.** **a**, **b** mRNA expression of DDAH isoforms in tissues of wild type and *Ddah1* knockout mice. **c** Representative Western blot image of protein expression in tissue lysates from wild type and *Ddah1* knockout mice. **d** Level of D7-labelled citrulline produced from the metabolism of D7-labelled ADMA by tissues of wild type and *Ddah1* knockout mice. Statistical analyses were performed by unpaired two-sided Mann–Whitney test for non-normally distributed data with comparisons to the wild-type group for each tissue type (*n* = 5 biological replicates). Outliers were removed by ROUT method (*Ddah1* KO liver, *Ddah1* KO kidney, *n* = 1 from *Ddah1* mRNA expression analysis). Data are presented as mean ± SE. **a** \**p* < 0.05 (*p* = 0.0159), \*\**p* < 0.01 (*p* = 0.0079), \*\*\**p* < 0.001 (heart *p* = 0.0003, lung *p* = 0.0006); **d** \*\**p* < 0.01 (*p* = 0.0079). Source data are provided as a Source Data file.

DDAH1-citrulline complex (PDB ID: 2JAI)[43]. Similarly, ADMA was positioned within the DDAH2 binding site by aligning to citrulline as a reference position using the DDAH1 X-ray structure (PDB ID: 2JAI)[43]. A steepest descents minimisation followed by a position restraint simulation for 250 ps was performed under a constant volume (NVT) ensemble. Constant pressure (NPT) equilibration was performed for 250 ps using weak coupling to maintain pressure isotropically at 1.0 bar at a temperature of 300 °K. A Parrinello-Rahman barostat was used to isotropically regulate pressure along with a velocity rescale thermostat to maintain temperature[78,79]. SETTLE and LINCS algorithms were used to constrain the bond lengths of water and solute, respectively[80,81]. Production molecular dynamics simulations were conducted for 100 ns without any restraints with the molecular trajectories saved every 100 ps.

### Expression and purification of recombinant human DDAH1, human DDAH2 and GST protein

The expression and purification of recombinant human DDAH1 (rDDAH1) and human DDAH2 (rDDAH2) protein was performed by DAPCEL Inc. (USA) (CDS: NM_012137 and NM_001303007; 2-285aa)[82]. Both DDAHs were expressed as N-terminal fusions with GST-tag using pGEX-4T-1 (GE Healthcare) vector in *E. coli* One Shot® BL21 (DE3) cells (Thermo Fisher Scientific, Waltham). The resulting recombinant proteins were purified by affinity chromatography on glutathione Sepharose beads. Recombinant GST (rGST) was expressed in the same manner. Cells were grown in Luria-Bertani media for 1 h at 37 °C in oscillation and expression was induced by 0.1 mM isopropyl-beta-D-thiogalactopyranoside (IPTG) for 8 h at 30 °C, until the exponential growth phase. Cells were chemically lysed using B-PER Bacterial

Protein Extraction Reagent (Thermo Fisher Scientific, Waltham) according to the manufacturer's recommendations with the addition of EDTA-free complete protease inhibitor (Roche). The lysate was centrifuged at 18,000 × *g* for 30 min at 4 °C and the supernatant was filtered. Proteins were purified using Glutathione Sepharose 4B beads (GE Healthcare, Uppsala) column according to the manufacturer's recommendations using gravity flow. The protein was then eluted from the beads by incubating the resin in reduced glutathione elution buffer (50 mM Tris-HCl, 150 mM NaCl, 50 mM reduced glutathione, pH 8.5) for 15 min at room temperature for a total of three elution steps. The protein was then purified of reduced glutathione through a 50 kDa size exclusion filter (Millipore) and reconstituted in 20 mM Tris-HCl, 150 mM NaCl, pH 8.5.

The intrinsic fluorescence spectrum of each protein was registered using a Jasco FP500 instrument with an excitation wavelength set at 280 nm at 1 mg/ml protein concentration in 50 mM Tris, 150 mM NaCl, pH 8.5.

### Microscale thermophoresis (MST)

DDAH1-GST, DDAH2-GST, and GST were fluorescently labelled at lysine residues with red dye NT-650-NHS supplied by NanoTemper Technologies (NanoTemper Technologies, GmbH, Munich, Germany), using 1:3 protein/dye ratio for DDAH2-GST and GST, whereas 1:6 protein/dye ratio for DDAH1-GST. Mixtures of proteins and NT-650-NHS fluorophore were incubated for 30 min at room temperature in the dark, according to the instructions of the manufacturer. For the labelling process, a buffer composed of 130 mM NaHCO₃, 50 mM NaCl, pH 8.2, enriched with 0.05% Tween20 was used. Unbound dye was then removed by size-exclusion chromatography with the running

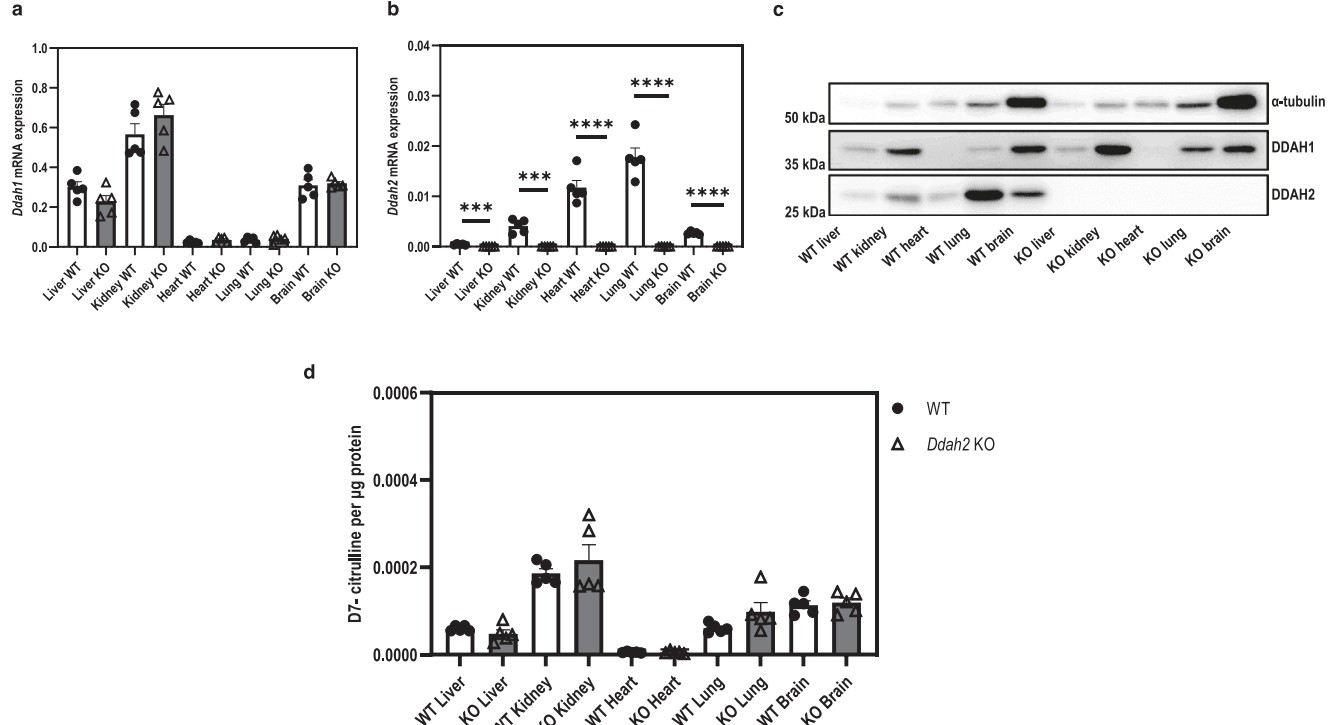

**Fig. 9 | DDAH activity assay of tissues from wild type and *Ddah2* knockout mice.** **a**, **b** mRNA expression of DDAH isoforms in tissues of wild type and *Ddah2* knockout mice. **c** Representative Western blot image of protein expression in tissue lysates from wild type and *Ddah2* knockout mice. **d** Level of D7-labelled citrulline produced from the metabolism of D7-labelled ADMA by tissues of wild type and *Ddah2* knockout mice. Statistical analysis was performed using two-sided unpaired *t*-test for normally distributed data (**a**, **b**) and unpaired two-sided Mann–Whitney test for non-normally distributed data (**d**), with comparisons to the wild-type group of each tissue type ($n = 5$ biologically replicates). Data are presented as mean ± SE. ***$p < 0.001$ ($p = 0.0001$), ****$p < 0.0001$. Source data are provided as a Source Data file.

buffer consisting of 50 mM Tris-HCl, pH 8.5, 0.05% Tween20. Concentrations of the proteins, NT-650-NHS fluorophore dye, and labelled proteins were assessed using Absorbance Spectroscopy (AS) with the Thermo Scientific™ NanoDrop™ One spectrophotometer (Thermo Fisher Scientific Inc., Waltham Massachusetts, USA). The degree of labelling (DOL) was determined as the ratio between the dye and protein concentration in the sample. Obtained DOL values were within the range of 0.5-1. Before running MST assays, background noise of labelled DDAH1-GST, DDAH2-GST, and GST was evaluated using: (1) 50 nM of labelled DDAH1-GST in a buffer composed of Tris-T, 1% DMSO and 5 mM of 1,10 phenanthroline (MW = 198.22 g/mol, Sigma-Aldrich Catalogue #320056), yielding a background noise value of 2.8 fluorescence unit; (2) 50 nM red of labelled DDAH2-GST in a buffer composed of Tris-T, 1% DMSO, and 5 mM of 1,10 phenanthroline, yielding a background noise value of 3.5 fluorescence unit; (3) 50 nM of labelled GST in a buffer composed of Tris-T, 1% DMSO, and 5 mM of 1,10 phenanthroline, yielding a background noise value of 0.9 fluorescence unit. Experiments were executed using setting LED power at 40% and medium MST power for labelled DDAH2-GST and labelled GST while setting LED power at 80% and medium MST power was used for labelled DDAH1-GST. Thermophoretic measurements were performed by titrating 50 nM of the labelled proteins with scalar concentration of ADMA. A total of 16 samples at decreasing ligand concentrations were thus prepared from an initial stock solution of 1 mM. The samples were incubated for 60 min at room temperature before loading in standard capillaries (NanoTemper Technologies, GmbH, Munich, Germany). The 16 capillary tubes were then submitted to a *Cap Scan* analysis of fluorescent emission at room temperature to verify whether each tube contained the same amount of labelled protein and the presence of protein sticking to capillary walls. Thermophoretic migrations were recorded using Monolith NT.115 instrument (NanoTemper

Technologies, GmbH, Munich, Germany) at the setting LED power at 40% and medium MST power for labelled DDAH2-GST and labelled GST while setting LED power at 80% and medium MST power was used for labelled DDAH1-GST. Recorded data were processed with Nano-Temper's *MO.Affinity Analysis v2.3* in *Default on Time mode (DoT)*, setting the *Hot Region* between 4/5 s. Confidence values (±) were indicated next to $K_d$ value for each of potential binders tested. Specifically, confidence values define the range where the $K_d$ falls with a 68% of certainty, as declared by NanoTemper Technologies. For each $K_d$ value, an amplitude value is assigned and is calculated as the difference between the unbound (baseline resulting from the lowest tested concentrations of ligand) and bound (plateau resulting from the highest tested concentrations of ligand) MST signals, which is then expressed as ‰Fnorm units. A given binding curve is deemed of high significance (i.e. presence of a strong interaction signal) if the amplitude value is three times higher than the background noise signal. The signal to noise ratio (S/N) is calculated to evaluate the quality of the collected binding data. It is calculated as the amplitude divided by the noise of the measurement. As a rule of thumb, signal-to-noise ratio higher than 5 suggests a good assay, whereas a ratio higher than 12 indicates an excellent assay.

**Enzymatic activity of recombinant DDAH1 and DDAH2 proteins**
ADMA (N^G,N^G-dimethylarginine dihydrochloride, MW = 275.18 g/mol) was supplied by Sigma-Aldrich (# D4268-50G). The additive reagent 1,10 phenanthroline (MW = 198.22 g/mol) was supplied by Sigma-Aldrich (#320056) in solid form dissolved in 100% DMSO to reach concentration stock of 1 M. A total of 3 μg protein (rDDAH1 or rDDAH2) was combined with 10 mM ADMA in 50 mM Tris, 5 mM 1,10-phenanthroline, 5 mM tris(2-carboxyethyl)phosphine (TCEP), pH 8.5 for 1 h at 37 °C. A colour developing reagent was prepared by mixing 1

part of 80 mM DAMO, 2 mM thiosemicarbazide with 3 parts of 17.35% (v/v) phosphoric acid, 33.7% (v/v) sulfuric acid, 0.765 mg/ml ammonium iron sulfate. Next, 600 µl of the colour developing reagent was added to each reaction mixture and the tubes were heated at 95–100 °C in a heating block for 15 min. The tubes were cooled at room temperature for 5 min. For the assay, 200 µl of each mixture was loaded on a 96-well plate and the absorbance was measured at 540 nm. Blank samples (i.e. containing all the assay components except the enzyme) were run and their absorbance was subtracted from that of each sample. The enzymatic activity was obtained by determining the amount of L-citrulline produced using a reference standard curve ran in parallel with the samples. GST protein was used as an additional control for the assay. Measurements were performed in at least three experiments unless noted otherwise.

## Cell culture

The human embryonic kidney cell line, HEK293T (#CRL-3216) and triple negative human breast cancer cell line, MDA-MB-231 (#CRM-HTB-26) were purchased from American Type Cell Culture (ATCC) and cultured in Dulbecco's Modified Eagle's Medium (DMEM) (Gibco) and DMEM with high glucose and sodium pyruvate from Gibco respectively, supplemented with 10% fetal bovine serum (FBS) and 1% Penicillin/Streptomycin (Gibco). Human umbilical vein endothelial cells (HUVECs) (Lonza, #C2157) from a single female donor were passaged in EBM endothelial cell basal medium supplemented with EGM-MV Single-Quots (Lonza) and grown on culture plates with bovine collagen (R&D) coating. Cells were trypsinised using 0.05% Trypsin-EDTA (Gibco) for regular passaging and grown on culture plates or flasks as an even monolayer in an incubator at 37 °C with 5% $CO_2$.

## Overexpression of human DDAH in HEK293T cells

Overexpression of the genes in the cell model was performed according to protocols published by Tommasi et al.[83] and Lewis et al.[84]. Briefly, human *DDAH1* and *DDAH2* coding sequences (CDS: NM_012137 and NM_001303007 respectively)[82] were cloned into the pEF-IRES(6) mammalian expression vector. Equal amounts of constructs (4 µg) were transfected into HEK293T cells using Lipofectamine2000 in OptiMEM (Gibco). The culture medium was supplemented with puromycin (1 mg/l) to generate the respective stable expression of recombinant human *DDAH*(s) in HEK293T cells.

## CRISPR/Cas9 DDAHs knockout generation in MDA-MB-231 cell line

Clustered regularly interspaced short palindromic repeat (CRISPR) and CRISPR associated 9 protein (Cas9) system was used to introduce a gene knockout in the cell line by non-homologous end joining method. Two CRISPR-Cas9 RNAs (crRNAs) each targeting the third coding exon of *DDAH1* gene (NM_012137.4)[82] and the first coding exon of *DDAH2* gene (NM_001303007.2)[82] with the highest on-target and off-target scores were selected based on the Geneious 8.1.6 software scoring system (https://www.geneious.com). The following crRNAs were ordered from Integrated DNA Technologies (IDT): DDAH1 5′-TGTCATGGAACATAGTGAGC-3′, 5′-TAATGGTGTTTACGCTTCAC-3′; DDAH2 5′-ACGGCCGTGTCGCCAAGCAG-3′, 5′-GGGACACGGCCCTAATCACG-3′. The guides were introduced into the cell by transfection together with the entire CRISPR/Cas9 complex. The *DDAH1 + 2* knockout cell line was generated by performing the *DDAH1* knockout in the *DDAH2* knockout MDA-MB-231 cell line.

**RNP transfection.** The crRNA and trans-activating crRNA (tracrRNA) (#1072532, IDT) were reconstituted to 100 µM with RNase/DNase free water. The tracrRNA (0.5 µl) was incubated with 1 µl of each crRNA for 5 min at room temperature. Cas9 enzyme (#1081060, IDT) (7.5 µg) was added to the tracrRNA-crRNA mixture and allowed to complex for 15 min at 37 °C forming the ribonucleoprotein (RNP) complex.

Lipofectamine™ CRISPRMAX Transfection Reagent from Invitrogen (#CMAX00003) was used to introduce the RNP complex into the cells. Cas9 Plus reagent (5 µl) was added to the RNP complex in a tube containing 125 µl OptiMEM medium while 7.5 µl of CRISPRMAX reagent was added to 125 µl OptiMEM medium in a separate tube. Both solutions were incubated at room temperature for 5 min. Following that, the diluted CRISPRMAX reagent was added to the RNP-Cas9 Plus reagent mixture and incubated at room temperature for 10 min. The RNP-transfection reagent complex was added to freshly trypsinised cells (200000 cells/well) resuspended in OptiMEM medium in a 6-well plate and incubated for 24 h at 37 °C. The cell culture medium was replaced with MDA-MB-231 culture medium after 24 h.

## Knockdown of DDAH in HUVECs

Short hairpin RNA (shRNA) for human DDAH and controls were from Open Biosystems (Huntsville, AL, USA). The DDAH1 shRNA targeting sequence was: 5′-ACACATTAGAAAGATCTGC-3′. The DDAH2 shRNA targeting sequences were shRNA-1: 5′-TATTGGTTCTGAGAGGGAG-3′; shRNA-2: 5′-CTACTTCCTATACTATCCT-3′. Lentiviruses of DDAH and control shRNAs were prepared in HEK293T cells transfected with targeted gene (pGIPZ-DDAH shRNA and pGIPZ-control shRNA, respectively), pGag.Pol, and pVSV-G encoding the cDNAs of the proteins that are required for virus packing. Then these lentiviruses were used to infect HUVECs, which were selected with puromycin (2 µg/ml) for 2 days and then used for experiments.

## Animals and tissue harvest

Global *Ddah1* deficient (*Ddah1*[−/−]) C57Bl/6J mice[35] were bred from heterozygous pair (*Ddah1*[+/−]) and genotyped using the following primers: wild-type allele forward 5′-AATCTGCACAGAAGGCCCTCAA-3′, reverse 5′-GGAGGATCCATTGTTACAAGCCCTTAACGC-3′; knockout allele forward 5′-TGCAGGTCGAGGGACCTAATAACT-3′, reverse 5′-AACCACACTGCTAGATGAAGTTCC-3′. Global *Ddah2* deficient (*Ddah2*[−/−]) mice line was purchased from Taconic (Model #TF0168). *Ddah2*[−/−] mice were bred from heterozygous pairs (*Ddah2*[+/−]) and genotyped using the following primers: knockout allele forward 5′-AAATGGCGTTACTTAAGCTAGCTTGC-3′; wild-type allele forward 5′-TTACCTCCTAGTACTCCATGCTCC-3′; reverse 5′-AAACAAAACAGCTTGGCTGGAAGG-3′. All animals were housed in a 12-h light dark cycle (lights switched on at 06:00) with food and water ad libitum, ambient temperature of 22–24 °C, and humidity of 45–46%. All efforts were made to reduce any suffering to the animals and the number of animals used. The mice were deeply anaesthetised with a concoction of 100 mg/kg ketamine and 10 mg/kg xylazine followed by perfusion with phosphate buffered saline (PBS) during tissue harvest (18–22 weeks) using protocols approved by the animal welfare committee of Technische Universität Dresden [Ethical permissions: DD25-5131/530/11 and DD24-9168.24-1/2014-2]. Mice grouping were as follows: *Ddah1*[−/−] colony− 3 males + 2 females; *Ddah2*[−/−] colony: wild-type−3 males + 2 females, knockout−4 males + 1 female. To avoid bias due to sex, each animal group consisted of mixed genders.

## Genomic DNA isolation, PCR, single sorting of cells and Sanger sequencing

Genomic DNA from the edited cells was isolated using the QIAamp DNA Mini Kit (#51304, Qiagen) according to manufacturer's instructions. Genotyping was done by polymerase chain reaction (PCR) using the respective primers, *DDAH1*, forward: 5′-TTTAGTGAAGCTGTTCTCTGTGGT-3′, reverse: 5′-ACATCTGCCAGGTGGTTGTAT-3′; *DDAH2*, forward: 5′-CAAAAGCTCAAAGGGAGCAC-3′, reverse: 5′-GGACTCCATCGACCTTAGGA-3′ (Biomers, Germany). The product was assessed by electrophoresis on 2% agarose gel. Cells were single sorted into a 96-well plate and grown from single colonies. Genotyping was repeated to confirm the deletion by sequencing (Microsynth

Seqlab, Germany). Sequencing primers were as follows: *DDAH1*, 5′-TTTAGTGAAGCTGTTCTCTGTGGT-3′; *DDAH2*, 5′-CAAAAGCT-CAAAGGGAGCAC-3′. Sequencing results were viewed using Benchling (https://benchling.com). Gene knockout clones were further verified by qPCR and Western blot.

## DDAH activity assay

Two approaches were used to measure DDAH enzymatic activity. Firstly, lysates from *DDAH1* and *DDAH2* overexpressed HEK293T cells were used to measure the rate of citrulline formation from the catabolism of ADMA by DDAH. The assay was performed according to the method described by ref. 85. The second approach used stable-isotope labelled ADMA (#DLM-7476-0, Cambridge Isotope Laboratories) for estimation of DDAH activity assay in cell and tissue lysates which was adapted and modified from protocols previously published[86–88] as shown below. The concentration of D7-labelled citrulline were assessed by HPLC-MS-MS and normalised to the total protein content in the corresponding samples.

**Cell lysate activity assay.** Cells were seeded in flasks and grown to 90–95% confluence. Cells were harvested by scraping in PBS, centrifuged at $200 \times g$ for 5 min, and resuspended in 1 ml of PBS buffer (pH adjusted to pH 6.4 using 1 M HCl) containing protease inhibitor and 1 mM PMSF. The cells were mechanically lysed using a sonicator, $3 \times 10$ s pulses at 40% Amplitude, and kept on ice in between pulses. The lysed cells were centrifuged for 10 min at $14,000 \times g$ at 4 °C and the supernatant (lysate) was collected. For the DDAH activity assay, 10 mM D7-labelled ADMA was added to obtain a final concentration of 100 μM labelled ADMA in 850 μl cell lysate. The lysate mixture was incubated at 37 °C for 24 h and collected by flash freezing in liquid nitrogen before analysis. The experiment was performed in triplicates for each group and repeated at least three times.

**Tissue lysate activity assay.** Mice tissues from *Ddah1⁻/⁻*, *Ddah2⁻/⁻* and their respective wild-type littermates (30–40 mg) were harvested and homogenised in 1 ml ice cold RIPA buffer containing protease inhibitor and 1 mM PMSF at 6500 rpm, twice for 20 s. The homogenised tissue was incubated on ice for 10 min and later centrifuged at $14,000 \times g$, 4 °C, for 10 min before collecting the lysate. For the assay, 88 μl of 10 mM D7-labelled ADMA was added to 100 μl of tissue lysate in 912 μl of PBS incubation buffer (adjusted to pH 8.0 with 1 M $H_3PO_4$) to reach a final concentration of 800 μM labelled ADMA. The reaction mixture was incubated at 37 °C for 1 h and later flash frozen in liquid nitrogen before analysis. Tissues were collected from liver, kidney, heart, lung, and brain of each animal ($n = 5$).

## RNA extraction, cDNA synthesis and qPCR

Cells (-1,000,000) were harvested and a total of 30 mg of frozen tissue was collected for RNA isolation, performed according to manufacturer's protocol of RNeasy Mini Kit (Qiagen). RNA concentration was measured with the Take3™ Micro-Volume Plate using the Synergy™ HTX Multi-Mode Microplate Reader. cDNA was synthesised using High Capacity cDNA Reverse Transcription Kit (Applied Biosystems) according to manufacturer's protocol, using between 500–2000 ng of RNA. qPCR was performed with SYBR Green Master Mix (Applied Biosystems) using Applied Biosystems 7300 machine Real-Time PCR system. Human cell line qPCR data were normalised to either human hypoxanthine phosphoribosyltransferase 1 (*HPRT*) or human beta-actin (*ACTB*) expression while all mice qPCR data were normalised to mouse beta-actin (*Actb*) expression (Supplementary Table 2).

## Western blot

Protein concentrations of cell and tissue lysates were measured according to manufacturer's protocol of Pierce™ BCA Protein Assay Kit (Thermo Scientific). For immunoblot analysis, 20 μg of lysate was diluted in Laemmli buffer (0.25 M Tris-HCl, 8% SDS, 40% glycerol, 0.2 mg/ml bromophenol blue, 20% β-mercaptoethanol) and distilled water. The samples were heat-denatured at 95 °C for 10 min and loaded onto 10% polyacrylamide gels and separated by SDS-PAGE under reducing conditions (20 min, 100 V followed by 60 min, 150 V). Proteins were transferred to polyvinylidene difluoride (PVDF) membranes using a tank blotting system from Bio-Rad (1 h, 100 V, 4 °C). Membrane was blocked in 5% blocking buffer (Tris buffered saline (TBS), 3% milk) for 1 h at room temperature and incubated overnight at 4 °C in primary antibody solution (TBS containing 0.2% Tween20 (TBST), 2% milk). Membrane was washed in TBST and incubated in secondary antibody solution (TBST, 2% milk) for 1 h at room temperature. Next, the membrane was washed in TBST and rinsed in TBS. The blotted proteins were detected using "Lumi-Light Western Blotting Substrate" (Roche) on a PeqLab Fusion Fx7 Imaging system and the data was analysed with ImageJ (version 1.53t). Antibodies used: anti-DDAH1 (monoclonal anti-DDAH1 (3H10)—previously described[89], 1:5000; ab180599—Abcam, 1:1000; PA5-52278—Thermo Fisher Scientific, 1:1000), anti-DDAH2 (ab232694, ab184166—Abcam, 1:1000; 14966-1AP—Proteintech, 1:1000; STJ28540—St John's Laboratory, 1:1000), anti-β-actin (3700S—Cell Signaling Technology, 1:5000), anti-β-actin-HRP (ab49900—Abcam, 1:30,000), anti-tubulin (T5168—Sigma-Aldrich, 1:5000), anti-rabbit-HRP (111-035-144—Jackson Immuno Research, 1:2000), anti-mouse-HRP (554002—Pharmingen, 1:2000).

## Statistical analysis

Data was organised using Microsoft 365 Excel while the statistical analysis was performed using GraphPad Prism 8 (GraphPad Software, San Diego, CA). Data were presented as mean ± SD for the in silico work and recombinant protein assays and as mean ± SE for the cell culture and in vivo work. Comparisons to the control group of each model were analysed by either unpaired Student's *t* test, Mann–Whitney test, Kruskal–Wallis test, or one-way analysis of variance (ANOVA) depending on the distribution of data. A *p* value of <0.05 was accepted as statistically significant (*$p < 0.05$, **$p < 0.01$, ***$p < 0.001$, ****$p < 0.0001$).

## Reporting summary

Further information on research design is available in the Nature Portfolio Reporting Summary linked to this article.

## Data availability

Molecular dynamics simulation initial and final configurations for DDAH1-X-ray, DDAH2-Model A (Pose A-C), and DDAH2-Model B are included in the Source data file. Publicly available data used in the study include the following: O95865, 2JAI, NM_012137.4, NM_001303007.2, NP_036269.1, NP_001289936.1. Source data are provided with this paper.

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

## Acknowledgements

The Genome Engineering Facility at Max Planck Institute, Dresden, helped design the crRNAs used in the CRISPR/Cas9 experiments. This research was undertaken with the assistance of resources from the National Computational Infrastructure (NCI), which is supported by the Australian Government. This work was partially supported by National Institutes of Health (HL148339 to Y.W. and HL140411 to D.M.), the German Academic Exchange Service (DAAD 57173983 to V.N.R.), the Technische Universität Dresden TransCampus Grant (TC2023_02_MED to R.N.R.), the German Research Foundation (DFG) grant (MA 3324/3-1 to R.M.), and German Heart Foundation/German Foundation of Heart Research (F/24/17 to R.N.R. and F/47/21 to N.J.). We thank Dr. Anton A. Komar from the Center for Gene Regulation in Health and Disease, Cleveland State University and Dr. Mirco Dindo from the Department of Medicine and Surgery, University of Perugia for their constructive advice on our work.

## Author contributions

R.N.R. envisioned the concept and coordinated the cooperation between the research centres. R.N.R., B.C., A.A.M., Y.W., V.N.R. A.M., S.T., N.J., R.M., S.M.B.-B., J.M.-L., and S.R.L. were involved in study design. P.C.N. performed the computational modelling and molecular dynamics simulation experiments. N.D.T. and D.R. expressed and purified recombinant proteins. E.B., L.R., and A.M. performed the MST studies. S.G. and B.C. assessed enzymatic activity of recombinant proteins. V.N.R., Y.W., S.T., R.S.A., R.M., X.H., and Y.C. were involved in the design and development of different cell and animal models. V.N.R., E.R., Y.W., R.S.A., S.C., X.H., Y.C., M.K., and T.S.-Y. performed cell and animal model validations. V.N.R., S.T., Y.W., R.S.A., J.M.-L., and S.M.B.-B. performed enzymatic activity assays in cell and animal models. V.N.R., P.C.N., R.N.R., B.C., A.A.M., A.M., N.J., R.S.A., L.R., E.B., S.G., N.D.T., D.R., S.R.L., S.T., J.M.-L., T.S.-Y., M.K., E.R., S.C., Y.C., X.H., N.B., P.M.S., N.W., S.R.B., D.M., S.M.B.-B., R.M., and Y.W. were involved in the data interpretation of all experiments. V.N.R., P.C.N., and R.N.R. wrote the initial draft of the manuscript. All authors contributed to further revisions of the manuscript draft and approved the final version.

## Funding

## Competing interests

The authors declare no competing interests.

## Additional information

[1]Department of Internal Medicine III, Technische Universität Dresden, Dresden, Germany. [2]Department of Clinical Pharmacology, College of Medicine and Public Health, Flinders University and Flinders Medical Centre, Bedford Park, Adelaide, SA, Australia. [3]Flinders Health and Medical Research Institute (FHMRI), College of Medicine and Public Health, Flinders University, Adelaide, SA, Australia. [4]Cancer Program, South Australian Health and Medical Research Institute (SAHMRI), University of Adelaide, Adelaide, SA, Australia. [5]Discipline of Medicine, Adelaide Medical School, University of Adelaide, Adelaide, SA, Australia. [6]Department of Biochemistry and Molecular Biology, Mayo Clinic College of Medicine and Science, Jacksonville, FL, USA. [7]Department of Pharmaceutical Sciences, University of Perugia, via del Liceo 1, Perugia, Italy. [8]Department of Medicine and Surgery, University of Perugia, P.le L. Sevari 1, Perugia, Italy. [9]DAPCEL, Inc., Cleveland, OH, USA. [10]Department of Internal Medicine, The University of Iowa Carver College of Medicine, Iowa City, IA, USA. [11]Institute of Clinical Pharmacology, Otto von Guericke University, Magdeburg, Germany. [12]Department of Nutritional Science, Faculty of Health and Welfare Science, Okayama Prefectural University, Okayama, Japan. [13]Department of Cardiovascular Medicine, Mayo Clinic College of Medicine and Science, Rochester, NY, USA. [14]Department of Physiology and Biophysics, University of Mississippi Medical Center, Jackson, MS, USA. [15]Institute of Molecular Medicine, Beijing University, Beijing, China. [16]Department of Psychiatry and Psychotherapy, University Hospital Carl Gustav Carus, Technische Universität Dresden, Dresden, Germany. [17]Department of Anesthesiology and Critical Care Medicine, University Hospital Dresden, Technische Universität Dresden, Dresden, Germany. [18]School of Cardiovascular and Metabolic Medicine and Sciences, Faculty of Life Sciences & Medicine, King's College London, London, UK. [19]Institute of Experimental and Clinical Pharmacology and Toxicology, Friedrich-Alexander-Universität Erlangen-Nürnberg, Erlangen, Germany. [20]FAU New - Research Center for New Bioactive Compounds, Friedrich-Alexander-Universität Erlangen-Nürnberg, Erlangen, Germany. [21] College of Medicine and Public Health, Flinders University and Flinders Medical Center, Adelaide, SA, Australia. [22]These authors contributed equally: Vinitha N. Ragavan, Pramod C. Nair. ✉e-mail: Roman.Rodionov@uniklinikum-dresden.de

