## [Peer Review File · Nature Communications]

REVIEWER COMMENTS

Reviewer #1 (Remarks to the Author):

The studies of Rodionov et al. clearly demonstrate that DDAH2 is not transforming ADMA to citrulline. However, I would not call this a paradigm shift, as the results from other studies like those described in ref. 34 have already mainly ruled out the involvement of DDAH2 in ADMA metabolism. I would agree with that the presented study clears away any doubts which others may have had.

The studies include all modern techniques (see the summary at the beginning of the discussion) to unequivocally prove that DDAH2 is not metabolizing ADMA. The question what its function is remains open. All methods are of high quality and described in a way that they can be repeated by others. All relevant references are included and discussed in a proper way.

So I recommend acceptance of the paper.

However, the title of the paper should be changed in a wording like " Definite proof: DDAH2 is not metabolizing ADMA ".

For a communication the manuscript seems too long. It could be easily shortened by deleting some repetitions.

Reviewer #2 (Remarks to the Author):

Overview: Asymmetric dimethylarginine (ADMA) is an endogenous homologue of L-arginine that inhibits nitric oxide synthase. ADMA is known to impair endothelial function, promote vascular inflammation, and accelerate atherogenesis in pre-clinical models, and is associated with cardiovascular morbidity and mortality. Dimethylarginine dimethylaminohydrolase 1 (DDAH1) is the major enzyme responsible for metabolism of ADMA. Its overexpression reduces ADMA levels, improves endothelial function, and reduces the progression of vascular disease in animal models. By contrast, there is controversy regarding the role of DDAH2 in ADMA metabolism.

This international consortium of research groups was assembled to understand the role of DDAH2 in ADMA metabolism. They have definitively addressed the question using *in silico*, *in vitro*, cell culture and murine models. The findings uniformly demonstrate that DDAH2 is incapable of metabolizing ADMA to citrulline.

The data is quite strong. The DDAH1 X-ray crystal structure was used as a template to model the 3D structure of DDAH2 using SWISS-MODEL (Model A), as well as Alpha Fold (Model B). Molecular docking and molecular dynamic studies indicated that ADMA is a strong ligand of DDAH1; in contrast, the formation of a stable ADMA-DDAH2 complex productive for the hydrolytic reaction is highly unfavorable. This in silico modeling was confirmed by thermophoresis studies indicating that ADMA bound with reasonable affinity to DDAH1, but the binding was less avid and unstable with DDAH2. Purified DDAH1, but not DDAH2, metabolized ADMA to citrulline in vitro. DDAH1 (but not DDAH2) overexpression in HEK293T cells increased ADMA metabolism to citrulline. A CRISPR-facilitated KO of DDAH1, but not DDAH2, abolished the ability of cells to metabolize ADMA to citrulline. Similar observations were made using lentiviral shRNA against DDAH1 or 2. Finally, DDAH1 KO animals, but not DDAH2 KO animals, exhibited a loss of tissue DDAH activity, as assessed by ADMA degradation studies.

The consortium of investigators have provided highly rigorous and reproducible data that DDAH2 does not metabolise ADMA.

Whereas this level of rigor is commendable, the significance of their observation is less so. The investigators state their work is important for “opening a new field for investigation of alternative, ADMA-independent functions of DDAH2”. But in fact, as they also state, such work is already ongoing, with multiple publications revealing that DDAH 2 regulates mitochondrial fission, angiogenesis, vascular remodelling, insulin secretion, and immune responses.

Minor comments:

“DDAH2 may prove to be a more “targetable” enzyme than DDAH1.” What is the evidence for this statement?

Line 126 and Line 133 p6 “complementary” is misspelled

Reviewer #3 (Remarks to the Author):

The authors describe a combined computational/experimental study to investigate the binding and metabolism of ADMA by the protein DDAH2. Using a combination of techniques it is found that, in contrast to the highly similar DDAH1, DDAH2 does not metabolize ADMA, which has substantial pharmaceutical and clinical implications. This is an important study and overall the results are convincing and the manuscript is well written. However, I have a number of concerns about the

computational part of the manuscript that should be addressed before I can recommend the work for publication.

First, the MD work would be much more convincing if a quantitative analysis of ADMA (un)binding would be shown, instead of the illustrative structure pictures and movies. For example, time traces of ADMA RMSD with respect to the protein could be used for this purpose.

Second, the statistical basis of the MD findings remain unclear. The text mentions "Multiple subsequent molecular dynamics simulations" for model A, but to judge the statistical significance, multiple traces for model A, model B, as well as for the positive control DDAH1 should be presented.

We sincerely appreciate the Reviewers' time, comments and suggestions. Herein, we address all comments in a point-by-point manner in the hope that after revision our manuscript will be acceptable for publication.

Reviewer #1 (Remarks to the Author):

The studies of Rodionov et al. clearly demonstrate that DDAH2 is not transforming ADMA to citrulline. However, I would not call this a paradigm shift, as the results from other studies like those described in ref. 34 have already mainly ruled out the involvement of DDAH2 in ADMA metabolism. I would agree with that the presented study clears away any doubts which others may have had. The studies include all modern techniques (see the summary at the beginning of the discussion) to unequivocally prove that DDAH2 is not metabolizing ADMA. The question what its function is remains open. All methods are of high quality and described in a way that they can be repeated by others. All relevant references are included and discussed in a proper way. So I recommend acceptance of the paper. However, the title of the paper should be changed in a wording like "Definite proof: DDAH2 is not metabolizing ADMA". For a communication the manuscript seems too long. It could be easily shortened by deleting some repetitions.

Reply:

We thank the reviewer for the positive remarks on our consortium manuscript. In agreement with the reviewer's suggestion, we have changed the title of the manuscript to "*Definitive proof: Dimethylarginine dimethylaminohydrolase 2 is not a dimethylarginine dimethylaminohydrolase*" (Lines 2 - 3). Also, as the reviewer has proposed, we have shortened the manuscript by deleting some repetitive material (Lines 54 - 55, Lines 179 - 180, Line 186, Lines 677 - 686, Lines 689 - 697).

Reviewer #2 (Remarks to the Author):

Overview: Asymmetric dimethylarginine (ADMA) is an endogenous homologue of L-arginine that inhibits nitric oxide synthase. ADMA is known to impair endothelial function, promote vascular inflammation, and accelerate atherogenesis in pre-clinical models, and is associated with cardiovascular morbidity and mortality. Dimethylarginine dimethylaminohydrolase 1 (DDAH1) is the major enzyme responsible for metabolism of ADMA. Its overexpression reduces ADMA levels, improves endothelial function, and reduces the progression of vascular disease in animal models. By contrast, there is controversy regarding the role of DDAH2 in ADMA metabolism. This international consortium of research groups was assembled to understand the role of DDAH2 in ADMA metabolism. They have definitively addressed the question using in silico, in vitro, cell culture and murine models. The findings uniformly demonstrate that DDAH2 is incapable of metabolizing ADMA to citrulline. The data is quite strong. The DDAH1 X-ray crystal structure was used as a template to model the 3D structure of DDAH2 using SWISS-MODEL (Model A), as well as Alpha Fold (Model B). Molecular docking and molecular dynamic studies indicated that ADMA is a strong ligand of DDAH1; in contrast, the formation of a stable ADMA-DDAH2 complex productive for the hydrolytic reaction is highly unfavourable. This in silico modelling was confirmed by thermophoresis studies indicating that ADMA bound with reasonable affinity to DDAH1, but the binding was less avid and unstable with DDAH2. Purified DDAH1, but not DDAH2, metabolized ADMA to citrulline in vitro. DDAH1 (but not DDAH2) overexpression in

HEK293T cells increased ADMA metabolism to citrulline. A CRISPR-facilitated KO of DDAH1, but not DDAH2, abolished the ability of cells to metabolize ADMA to citrulline. Similar observations were made using lentiviral shRNA against DDAH1 or 2. Finally, DDAH1 KO animals, but not DDAH2 KO animals, exhibited a loss of tissue DDAH activity, as assessed by ADMA degradation studies.

The consortium of investigators has provided highly rigorous and reproducible data that DDAH2 does not metabolise ADMA. Whereas this level of rigor is commendable, the significance of their observation is less so. The investigators state their work is important for “opening a new field for investigation of alternative, ADMA-independent functions of DDAH2”. But in fact, as they also state, such work is already ongoing, with multiple publications revealing that DDAH 2 regulates mitochondrial fission, angiogenesis, vascular remodelling, insulin secretion, and immune responses.

Minor comments:

“DDAH2 may prove to be a more “targetable” enzyme than DDAH1.” What is the evidence for this statement?

Line 126 and Line 133 p6 “complementary” is misspelled.

Reply:

We thank the Reviewer for the positive comments, recognising our joint effort to resolve the controversy regarding the role of DDAH2 in ADMA metabolism. We completely agree with the reviewer that studies of DDAH2 involvement in different physiological and pathophysiological processes are already ongoing and producing exciting results, which makes this research area highly important for the broad audience. However, almost all, if not all of those ongoing studies, instead of investigation of the molecular mechanisms behind the observed biological effects of DDAH2, rather refer to the initially reported enzymatic activity of DDAH2 towards ADMA as the supposedly “well-known molecular mechanism of action of DDAH2” with subsequent reference to the multiple established biological effects of the ADMA/nitric oxide/superoxide system. This 20 years old misconception regarding the molecular mechanisms behind the biological effects of DDAH2 being supposedly well-established and ADMA-mediated is currently the major road block in the field and the focus of the current consortium study.

We have removed the statement that DDAH2 may prove to be a more targetable enzyme than DDAH1 (Line 57 and Line 124).

We appreciate the thorough inspection of our manuscript by the Reviewer and the detection of the misspelled word which was an oversight on our part (Line 132 and Line 139). We have corrected it and also double-checked the spelling in the rest of the manuscript.

Reviewer #3 (Remarks to the Author):

The authors describe a combined computational/experimental study to investigate the binding and metabolism of ADMA by the protein DDAH2. Using a combination of techniques, it is found that, in contrast to the highly similar DDAH1, DDAH2 does not metabolize ADMA, which has substantial pharmaceutical and clinical implications. This is an important study and overall the results are convincing and the manuscript is well written. However, I have a number of concerns

about the computational part of the manuscript that should be addressed before I can recommend the work for publication.

First, the MD work would be much more convincing if a quantitative analysis of ADMA (un)binding would be shown, instead of the illustrative structure pictures and movies. For example, time traces of ADMA RMSD with respect to the protein could be used for this purpose. Second, the statistical basis of the MD findings remains unclear. The text mentions “Multiple subsequent molecular dynamics simulations” for model A, but to judge the statistical significance, multiple traces for model A, model B, as well as for the positive control DDAH1 should be presented.

Reply:

We thank the Reviewer for the positive feedback on our paper and for highlighting the important point on how to further improve the quality of the manuscript. As the Reviewer suggested, we have incorporated the quantitative data of ADMA “(un)binding” using root mean square deviation (RMSD) analysis of ADMA with respect to the DDAH1 and DDAH2 structures based on five independent simulations for each individual DDAH structure (Fig. 2) (Lines 1006 – 1014). The graphic representation of the RMSD analysis of ADMA in the DDAH1 structure (positive control) is shown in Fig. 2e, while RMSD analysis of ADMA in DDAH2 Models A and B are shown in Fig. 2f, and 2g, correspondingly. The mean RMSD of ADMA in the DDAH1 structure (positive control) was 1.53 ± 0.24 (Line 181). In contrast, the mean RMSD of ADMA in DDAH2 Models A and B were 5.23 ± 2.23 and 3.80 ± 0.89 (p-values of 0.000077 and 0.002179, correspondingly, vs. the ADMA-DDAH1 RMSD, using one-way ANOVA (Lines 190 - 191, Lines 194 - 195) (Supplementary Table 1). We also have included data from the molecular dynamics simulations of ADMA in DDAH2 Model A in three different binding poses as additional supplementary information (Supplementary Fig. 5).

REVIEWERS' COMMENTS

Reviewer #3 (Remarks to the Author):

The authors have satisfactorily addressed my concerns.